# Explicit simulation of microbial transport with a dual-permeability, two-site kinetic deposition formulation using the integrated surface-subsurface hydrological model HydroGeoSphere

Friederike Currle<sup>1</sup>, René Therrien<sup>2</sup>, Oliver S. Schilling<sup>1,3</sup>

<sup>1</sup>Hydrogeology, Department of Environmental Sciences, University of Basel, Basel, CH-4056, Switzerland

<sup>2</sup>Department of Geology and Geological Engineering, Université Laval, Québec, QC, Canada

<sup>3</sup>Eawag, Swiss Federal Institute of Aquatic Science and Technology, Dübendorf, CH-8600, Switzerland

Correspondence to: Friederike Currle (friederike.currle@unibas.ch)

**Abstract.** Assessing the transport behavior of microbes in surface water-groundwater systems is important to prevent contamination of drinking water resources by pathogens. While wellhead protection area (WHPA) delineation is predominantly based on dye injection tests and advective transport modeling, size exclusion of colloid-sized microbes from the smaller and usually less conductive pore spaces causes a faster breakthrough and thus faster apparent transport of microbes compared to that of solutes. To provide a tool for better assessment of the differences between solute and microbial transport in surface water-groundwater systems, we here present the implementation of a dual-permeability, two-site kinetic deposition formulation for microbial transport in the integrated surface-subsurface hydrological model HydroGeoSphere (HGS). The implementation considers attachment, detachment and inactivation of microbes in both permeability regions and allows for multispecies transport. The dual-permeability, two-site kinetic deposition implementation in HGS was verified against an analytical solution for dual-permeability colloid transport. The suitability of the model for microbial transport in integrated surface-subsurface hydrological settings at the wellfield or small headwater catchment scale is demonstrated by two illustrative examples. The first example is a benchmark for integrated rainfall-runoff and streamflow generation modeling to which we added microbial transport from a conceptual manure application, demonstrating the novelty of explicit and coupled microbial and solute transport simulations in an integrated surface-subsurface hydrological scenario. The second example is a multitracer flow and transport study of an idealized alluvial riverbank filtration site, in which we simulate in parallel the transport of reactive microbes, conservative <sup>4</sup>He, and reactive <sup>222</sup>Rn, demonstrating the assessment of mixing ratios, tracer breakthrough curves and travel times in an integrated manner via multiple approaches. The developed simulation tool represents the first integrated surface-subsurface hydrological simulator for reactive solute and microbial transport, and marks an important advancement to unlock and quantify governing microbial transport processes in coupled surface water-groundwater settings. It enables meaningful WHPA delineation and risk assessments of riverbank filtration sites with respect to microbial contamination even under extreme hydrological and microbial stress situations such as flood events.

# 30 1 Introduction

Riverbank filtration is a popular and widely used method for drinking water production, as the filtration of river water through riverbanks and alluvial sediments naturally removes many microbial and chemical contaminants (Weiss et al., 2005; Hollender et al., 2018; Ray, 2002). To ensure the quality of the pumped filtrate, riverbank filtration wells are usually installed at distances from the river that allow for sufficient groundwater travel distance and time. Analogously to how wellhead protection areas (WHPA) or capture zones are delineated, these distances are typically identified based purely on hydraulic analyses, advective transport modeling, or under additional consideration of dye injection tests that allow for the delineation of time-of-travel WHPA (Molson and Frind, 2012; Paradis et al., 2007). However, especially after heavy rain and discharge events, groundwater pumped from riverbank filtration systems can contain high microbial loads, in spite of the natural filtration process (Derx et al., 2013; Brand and Wülser, 2018). Thus, understanding the transport of microbes in river-aquifer systems is crucial to prevent contamination of drinking water by pathogens, particularly in the widely employed riverbank filtration context (Bradford et al., 2017; Tufenkji et al., 2002; Ferguson et al., 2003).

In contrast to solutes, microbes are orders of magnitudes larger, with a typical size of 0.1 nm for simple ions compared to 0.1 to 750 µm for bacteria and other prokaryotes. Microbes are classified as colloids and are suspended rather than dissolved in water. Microbes can thus get trapped in pores too small for them to flow through - a process called 'straining'. Due to this size exclusion, microbial transport is limited to the larger and, usually, more conductive pores (Bradford et al., 2003), which is why microbes appear to travel faster than solutes, showing an earlier breakthrough (Hunt and Johnson, 2017) and posing an increased risk for drinking water contamination (Bradford and Harvey, 2017). WHPA delineated based on advective solute transport are therefore not representative for microbial transport pathways. Unfortunately, unlike dissolved chemical contaminants, for which chronic exposure is typically of most concern, a very small acute exposure to waterborne pathogens can cause illness (Hunt and Johnson, 2017; Sasidharan, 2016). Therefore, it is the first arrival of pathogens rather than that of the peak or the centre of mass that is the most relevant waterborne transport health risk to be considered in drinking water protection (Hunt and Johnson, 2017).

The transport and fate of microbes in the subsurface is mainly governed by advection, diffusion and dispersion, interaction with the grain surfaces, size exclusion, as well as growth and inactivation by grazing or death (Tufenkji, 2007; Bradford et al., 2014). Several different mathematical approaches and models to conceptualize the transport of these bio-colloids have been developed (Bradford et al., 2014). A conventional approach is the classical colloid filtration theory (CFT) (Yao et al., 1971), which is a form of the advection-dispersion-equation enhanced by a single kinetic attachment coefficient. This kinetic attachment coefficient is an upscaled rate coefficient considering both mass transport from the pore space to the grain surfaces as well as the attachment efficiency to the surface. Classical CFT assumes irreversible attachment of colloids to the sediment grains (Tufenkji, 2007; Ryan et al., 1999; Schijven and Hassanizadeh, 2000). For attractive forces between colloid and collector (i.e. sediment grains), which correspond to the so-called favorable conditions for attachment, various column experiments

confirmed that the single-rate description of the classical CFT allowing fast and irreversible attachment is in good agreement with observations of the colloid concentrations in the liquid phase and, if available, also the retained colloid concentrations (Li et al., 2004; Li et al., 2005; Tufenkji and Elimelech, 2004, 2005). However, classical CFT breaks down under unfavorable conditions, hence, in the presence of repulsive forces between colloid and collector (Tufenkji and Elimelech, 2004, 2005; Li et al., 2005). Colloid attachment under unfavorable conditions is either fast and reversible for colloids that are retained in the secondary energy minimum (i.e. shallow energy minimum for weak colloid attraction), or slow and irreversible for colloids that overcome the repulsive energy barrier and reach the primary energy minimum (Tufenkji, 2007). Attachment conditions of microbes in the environment are usually unfavorable, as surfaces of both microorganisms and sediment grains are typically negatively charged (Loveland et al., 1996; Ryan et al., 1999; Molnar et al., 2015; Bradford et al., 2009) and, consequently, the classical CFT is not appropriate to describe microbial transport under environmental conditions.

70

A common enhancement of the classical CFT is a one-site kinetic sorption model with additional first-order detachment and/or inactivation-rates (Schijven et al., 2000; Zhang et al., 2001; Dong et al., 2002; Blanford et al., 2005; Zhang et al., 2018; Oudega et al., 2021; Knabe et al., 2021). Beyond that, Bradford et al. (2003) developed an irreversible first-order straining term to incorporate straining in addition to a first-order attachment and detachment. The dual-deposition mode by Tufenkji and Elimelech (2004) comprises a bimodal distribution of the deposition rate in the classical CFT to consider both fast and slow deposition under favorable and unfavorable attachment conditions. Furthermore, two-site kinetic sorption models have been shown to reproduce the breakthrough of bio-colloids both in column as well as field experiments. Schijven and Šimůnek (2002), for example, showed that the shape of the breakthrough curves of a bacteriophage tracer experiment at a managed aquifer recharge site could be fitted well using a 1D two-site model. Sasidharan et al. (2018) outlined that a two-site kinetic retention model with different solid-phase inactivation rates describes accurately the observed tailing of a virus breakthrough curve of a column experiment. While all these models addressed colloid attachment, detachment, inactivation, or straining, they did not consider parallel simulation of the different transport characteristics of colloids and solutes.

Making another step forward, Bradford et al. (2009) presented a dual-permeability model (also known as a dual-region model) with two kinetic deposition sites to account for different colloid retention mechanisms. Initially applied to study preferential flow and transport in structured porous media and fractured rocks (Gerke and Van Genuchten, 1993), Bradford et al. (2009) reinterpreted the dual-permeability model to account for facilitated colloid transport in porous media. While the mathematical formulation remains similar to its traditional use, the reinterpretation addresses a different physical system in which colloid attachment is influenced by small-scale velocity variations within the porous medium. Slow and irreversible colloid attachment is considered to occur in the small-pore, low-velocity regions of the porous media, whereas fast and potentially reversible retention occurs in the large-pore, high-velocity region. The dual-permeability approach was able to reproduce the breakthrough curve and the retention profile of a monodispersed colloid suspension in a packed column under highly unfavorable conditions and performed better compared to a single-region attachment-detachment model (Bradford et al., 2009). In addition to the application for colloid transport under unfavorable conditions, the dual-permeability model formulation can be adjusted to completely favorable or not clearly defined and mixed favorable - unfavorable conditions

(Bradford et al., 2014), making it an ideal and flexible approach for natural heterogeneous porous media. Furthermore, the dual-permeability model enables the co-simulation of fast microbial transport in the high-permeability pore-space and slower bulk solute transport in both the high- and low-permeability pore spaces. The dual-permeability, two-site kinetic deposition model is therefore a promising approach to co-simulate microbial and solute transport under environmental conditions, which is required for the characterization of relevant processes in river-groundwater systems.







Bradford et al. (2009) performed simulations according to the dual-permeability formulation with two kinetic deposition sites using the finite-element, variably saturated subsurface flow and transport model HYDRUS 1D (Šimůnek and Van Genuchten, 2008). HYDRUS 1D, as well as its 2D and 3D extensions HYDRUS 2D and HYDRUS 3D (Šimůnek et al., 2013; Šimůnek et al., 2016), are well suited to study flow and transport phenomena in the variably saturated zone. However, to study the transport phenomena on the surface, in the subsurface and, in particular, across the river-aquifer interface, integrated surface-subsurface hydrological models (ISSHM) are needed. ISSHM simulate both the surface as well as the variably and fully saturated subsurface hydrological environments in a fully coupled manner and thereby eliminate the necessity to parametrize the surface domain as external boundary conditions (BC) (Brunner et al., 2017; Schilling et al., 2019; Maxwell et al., 2014; Partington et al., 2017; Delottier et al., 2024). Examples of ISSHM include HydroGeoSphere (Aquanty Inc., 2024; Therrien and Sudicky, 1996; Brunner and Simmons, 2012), Cast3M (Weill et al., 2009), CATHY (CATchment HYdrology) (Bixio et al., 2002; Camporese et al., 2010), or ParFlow (Maxwell et al., 2024; Beisman et al., 2015).

So far, none of these ISSHM included the possibility to study reactive colloid transport including interactions with the grain surfaces in a dual-permeability system. To fill this gap, we present here the first implementation and verification of a dualpermeability, two-site kinetic deposition flow and transport formulation, combined with first-order decay terms in both permeability regions, in an integrated surface-subsurface flow and transport model. The new implementation was first announced in the HydroGeoSphere release notes rev. 2582 and the final implementation as presented in this study has been available in HydroGeoSphere from rev. 2699 onwards. The suitability of the approach to simulate microbial transport at the wellfield scale in integrated surface-subsurface hydrological systems is highlighted by two illustrative examples. The first illustrative example is based on an existing rainfall-runoff benchmark that is modified to simulate streamflow generation from groundwater exfiltration. The remainder of the paper also refers to groundwater exfiltration as return flow. This first example demonstrates the potential of the new implemented approach to study microbial transport in river-aquifer systems with varying in- and exfiltrating conditions. The second illustrative example uses a riverbank filtration benchmark model and simulates microbial transport in tandem with the reactive transport of the dissolved natural radiogenic and radioactive gas tracers <sup>4</sup>He and <sup>222</sup>Rn, respectively. The latter have been shown to be suitable tracers to estimate groundwater mixing ratios and residence times of freshly infiltrated river water in riverbank filtration wellfields (Popp et al., 2021). An artificial <sup>4</sup>He injection pulse in the river water is also simulated based on recent work by Blanc et al. (2024) who demonstrate that, for such an injection, <sup>4</sup>He is a conservative tracer as efficient as a dye.

# 2 Microbial transport in HydroGeoSphere

# 130 2.1 Coupled surface water and dual-permeability subsurface flow

HydroGeoSphere (HGS) is an integrated surface-subsurface hydrological model capable of simulating surface water and variably-saturated subsurface flow as well as heat and (reactive) mass transport. It has been employed to study many aspects of river-aquifer interactions, such as the influence of riverbed heterogeneity (Tang et al., 2017) and the impact of catchment scale river-aquifer interactions on groundwater levels (Delottier et al., 2024). It has also be used to improve model calibration by including river-aquifer exchange fluxes (Schilling et al., 2022).

Variably-saturated flow in the subsurface in HGS is described by a modified form of Richards' equation to account for specific storage in the saturated zone (Therrien and Sudicky, 1996). In the dual-permeability configuration, the subsurface is subdivided into a high-permeability and a low-permeability region. Flow is described for both regions individually using a modified form of Richards' equation (Gerke and Van Genuchten, 1993):

$$w_{\rm h} \frac{\partial(\theta_{\rm sh} S_{\rm wh})}{\partial t} = -\nabla \cdot (w_{\rm h} q_{\rm h}) + \Gamma_{\rm oh} + \Gamma_{\rm hl} \pm Q_{\rm h} \tag{1}$$

$$w_{l} \frac{\partial(\theta_{sl}S_{wl})}{\partial t} = -\nabla \cdot (w_{l}\boldsymbol{q}_{l}) + \Gamma_{ol} - \Gamma_{hl} \pm Q_{l}$$
(2)

where subscripts h and l refer to the high-permeability and low-permeability regions, respectively. The volume fraction of each region is given by w [-] with  $w_h + w_l = 1$ ,  $\theta$  [-] is the saturated water content, which is equal to porosity, S [-] is the relative water saturation, and  $\Gamma_0$  [L<sup>3</sup> L<sup>-3</sup> T<sup>-1</sup>] is the volumetric fluid exchange rate between the surface and each subsurface region, and Q [T<sup>-1</sup>] are sources and sinks as, e.g., defined by boundary conditions.

The fluid fluxes q [L T<sup>-1</sup>] in the high- and low-permeability subsurface regions are defined as:

$$\mathbf{q_h} = -\mathbf{K_h} \cdot k_{rh} \, \nabla(\psi_h + z) \tag{3}$$

$$\mathbf{q}_{\mathbf{l}} = -\mathbf{K}_{\mathbf{l}} \cdot k_{r} \nabla (\psi_{\mathbf{l}} + z) \tag{4}$$

Where **K** [L T<sup>-1</sup>] is the hydraulic conductivity tensor,  $k_r$  [-] is the relative permeability, which is a function of water saturation,  $\psi$  [L] is pressure head, and z [L] is the elevation head.

The water exchange term between the high- and low-permeability regions  $\Gamma_{hl}$  [L<sup>3</sup> L<sup>-3</sup> T<sup>-1</sup>] is defined as:

$$\Gamma_{\rm hl} = \alpha_w K_a k_{ra} (\psi_{\rm l} - \psi_{\rm h}) \tag{5}$$

where  $\alpha_w$  [L<sup>-2</sup>] is a first-order fluid transfer coefficient,  $K_a$  [L T<sup>-1</sup>] is the hydraulic conductivity at the interface between the two regions, and  $k_{ra}$  [-] is the relative permeability of the interface.

The total flux  $q_t$  in the subsurface is then given by:

$$q_{t} = w_{h}q_{h} + w_{l}q_{l} \tag{6}$$

A similar relationship holds to define the total hydraulic conductivity and total water saturation in the subsurface.



Surface flow in HGS is described by the following depth-averaged diffusion wave approximation:

$$\frac{\partial \phi_0 h_0}{\partial t} = -\nabla \cdot (d_0 \mathbf{q_0}) - d_0 \Gamma_0 \pm Q_0 \tag{7}$$

where  $h_0$  [L] is the surface water level,  $d_0$  [L] the surface water depth and  $d_0\Gamma_0$  [L T<sup>-1</sup>] represents flow between the surface and subsurface domains. The volumetric flow rate per unit area  $Q_0$  [L T<sup>-1</sup>] accounts for external sources and sinks and  $\phi_0$  [-] is the porosity of the surface domain which also considers rill storage and obstruction storage exclusion.

The 2D-fluid flux  $q_0$  [L T<sup>-1</sup>] is given by:






$$\mathbf{q}_{0} = -\mathbf{K}_{0} \cdot k_{r_{0}} \nabla (h_{0}) \tag{8}$$

where  $\mathbf{K_0}$  [L T<sup>-1</sup>] represents the surface conductance and  $k_{ro}$  [-] is a factor to reduce the conductance up to the obstruction height.

Coupling between the surface and the subsurface domain is achieved via a dual-node implementation. In general, the surface-subsurface exchange term is a first-order coupling term that depends on the head differences between the surface and subsurface domains over a user defined coupling length for the exchange. For a dual-permeability subsurface, the surface-subsurface exchange term is the sum of individual exchanges between the surface domain and the high- and low-permeability subsurface regions, respectively, and is defined as:

$$d_{o}\Gamma_{o} = w_{h} \frac{k_{rh}K_{zh}}{l_{erch}} (h_{h} - h_{o}) + w_{l} \frac{k_{rl}K_{zl}}{l_{erch}} (h_{l} - h_{o})$$
(9)

where  $h_h$  and  $h_l$  [L] are the hydraulic heads of the high- and low-permeability subsurface regions, and  $K_{zh}$  and  $K_{zl}$  [L T<sup>-1</sup>] the vertical saturated hydraulic conductivities.  $l_{exch}$  [L] is the coupling length between the surface and each of the two subsurface regions.

#### 180 2.2 Dual-permeability mass transport with two-site kinetic deposition mode

As outlined above, classical (reactive) solute transport or colloid filtration theory are not suitable for describing microbial transport in groundwater, but a modified colloid filtration approach that is conceptualized as a dual-permeability domain with two-site kinetic deposition mode has been shown to be suitable for the simulation of microbial transport at the wellfield scale. Here, we implemented the governing equations for modeling microbial transport in dual-permeability systems introduced by Bradford et al. (2009) into HGS. In this formulation, the pore space of the subsurface is divided into two regions and, in each region, colloids can exist both in the liquid phase and attached to the solid phase. The transfer of colloids between the liquid to solid phases is described through first-order kinetic terms, representing attachment and detachment. Bradford et al. (2009) only simulated colloid transport at the column scale. We extended the model by adding first-order sink terms in the liquid and solid phases to account for microbial inactivation relevant to the wellfield or catchment scale. The governing equations for the liquid concentrations in the high- and low-permeability subsurface regions are:

$$\frac{\partial(\theta_{\rm sh}S_{wh}c_{\rm h})}{\partial t} = -\nabla J_{\rm h} + \frac{\Omega_{ex}}{w_{\rm h}} - \theta_{\rm sh}S_{wh}k_{atth}c_{\rm h} + \rho_{bh}k_{deth}s_{\rm h} - \lambda_{\rm h}\theta_{\rm sh}S_{wh}c_{\rm h} + \Omega_{\rm oh}$$

$$\tag{10}$$

$$\frac{\partial(\theta_{sl}S_{wl}c_l)}{\partial t} = -\nabla J_l - \frac{\Omega_{ex}}{w_l} - \theta_{sl}S_{wl}k_{attl}c_l + \rho_{bl}k_{detl}S_l - \lambda_l\theta_{sl}S_{wl}c_l + \Omega_{ol}$$
(11)

Where c [N<sub>c</sub> L<sup>-3</sup>] is the liquid phase concentration and N<sub>c</sub> the number of colloids,  $k_{att}$  [T<sup>-1</sup>] the first-order attachment rate coefficients,  $k_{det}$  [T<sup>-1</sup>] the first-order detachment rate coefficient,  $\lambda$  [T<sup>-1</sup>] is the first-order sink term in the liquid phase, and J [N<sub>c</sub> L<sup>-2</sup> T<sup>-1</sup>] the sum of the advective and dispersive fluxes.  $\Omega_{oh}$  and  $\Omega_{ol}$  [N<sub>c</sub> L<sup>-3</sup> T<sup>-1</sup>] are the mass exchange terms between the surface domain and the high- and low-permeability subsurface region, respectively. The term  $\Omega_{ex}$  [N<sub>c</sub> L<sup>-3</sup> T<sup>-1</sup>] describes the mass exchange in the liquid phase between the two subsurface regions and is given by:

$$\Omega_{ex} = \omega_{ex} \, W_h \Theta_{sh} S_{wh} (c_l - c_h) \tag{12}$$

Where the coefficient  $\omega_{ex}$  [T<sup>-1</sup>] accounts for colloid transfer from the high-permeability liquid phase to the low-permeability liquid phase,

The solid concentrations in the high- and low-permeability subsurface regions are defined as:



$$\frac{\partial(\rho_{bh}s_h)}{\partial t} = \Theta_{sh}S_{wh}k_{atth}c_h - \rho_{bh}k_{deth}s_h - \frac{\rho_{bh}k_ts_h}{w_h} - \lambda_{sh}\rho_{bh}s_h$$
(13)

$$\frac{\partial(\rho_{bl}s_l)}{\partial t} = \Theta_{sl}S_{wl}k_{attl}c_l - \rho_{bl}k_{detl}s_l + \frac{\rho_{bl}k_ts_l}{w_l} - \lambda_{sl}\rho_{bl}s_l$$
(14)

Where s [N<sub>c</sub> M<sup>-1</sup>] is the solid phase concentration,  $\rho_b$  the bulk density [M L<sup>-3</sup>],  $\lambda_s$  [T<sup>-1</sup>] is the decay term in the solid phase. 200 while  $k_t$  [T<sup>-1</sup>] is a coefficient for the colloid exchange from the solid phase in the high-permeability region to the solid phase in low-permeability region, driven by rolling or sliding of microbes on solid surfaces due to hydrodynamic forces (Bradford et al., 2009; Molnar et al., 2015).,

The irreversible process of microbial straining in the smaller pore space can be incorporated either by assuming irreversible attachment in the low-permeability region by setting  $k_{det1} = 0$ , or, in settings of reversible attachment-detachment conditions in both the high- and low-permeability regions, by utilizing the first-order sink terms as effective parameters for inactivation and straining. Both approaches can eliminate microbes permanently from the liquid phase, but only by considering the elimination terms, we can address irreversible straining alongside reversible attachment-detachment in both permeability zones as two distinct processes.

The total flux concentration of the two liquid phases in the subsurface  $c_t$  [N<sub>c</sub> L<sup>-3</sup>] is:

$$c_{t} = \frac{w_{h}q_{h}c_{h} + w_{l}q_{l}c_{l}}{w_{h}q_{h} + w_{l}q_{l}}$$
(15)

The total solid phase concentration in the subsurface  $s_t$  [N<sub>c</sub> M<sup>-1</sup>] is given by:

$$s_{t} = \frac{w_{h}\rho_{bh}s_{h} + w_{l}\rho_{bl}s_{l}}{w_{h}\rho_{bh} + w_{l}\rho_{bl}}$$
(16)

# 2.3 Coupled surface-subsurface solute transport

Solute transport in the surface domain is defined in HGS as follows:

$$\frac{\partial (\phi_0 h_0 c_0)}{\partial t} = -\overline{\overline{\nabla}} \cdot (q_0 c_0 - \mathbf{D}_0 \phi_0 h_0 \overline{\nabla} c_0) - \phi_0 h_0 \lambda_0 c_0 - d_0 \Omega_0 \tag{17}$$

where  $c_0$  [M L<sup>-3</sup>] is the concentration in the surface water domain,  $\overline{\nabla}$  is the depth integrated gradient operator,  $\mathbf{D_0}$  [L<sup>2</sup> T<sup>-1</sup>] is the hydrodynamic dispersion tensor, and  $\lambda_0$  [T<sup>-1</sup>] is the first-order decay constant in the surface domain.  $\Omega_0$  [M L<sup>-3</sup> T<sup>-1</sup>] represents the mass exchange flux between the surface and subsurface domain.

In a surface-subsurface HGS model setup coupled with the dual-node approach, a first-order term is used to transfer mass between the surface and subsurface domains. In a dual-permeability subsurface system, the total surface-subsurface mass exchange is the sum of the exchange between the surface and each of the two subsurface regions, here the high- and low-permeability regions as introduced in the previous section. The total surface – subsurface mass exchange  $\Omega_0$  is conceptualized as:

$$d_{o}\Omega_{o} = d_{o}\Omega_{oh} + d_{o}\Omega_{ol} = d_{o}\Gamma_{oh}c_{upsh} + \left(\frac{|d_{o}\Gamma_{oh}|\alpha_{c} + \theta_{sh}S_{wh}\tau_{h}D_{free}}{l_{exch}d_{o}}\right)(c_{h} - c_{o})$$
$$+d_{o}\Gamma_{ol}c_{upsl} + \left(\frac{|d_{o}\Gamma_{ol}|\alpha_{c} + \theta_{sl}S_{wl}\tau_{l}D_{free}}{l_{exch}d_{o}}\right)(c_{l} - c_{o})$$
(18)

Where  $\alpha_c$  [L] is the coupling dispersivity between the surface and the subsurface,  $D_{free}$  [L<sup>2</sup> T<sup>-1</sup>] the free-solution diffusion coefficient and  $\tau$  [-] the tortuosity.  $c_{upsh}$  and  $c_{upsl}$  [M L<sup>-3</sup>] are the upstream concentration terms for the exchange between the surface and the high- and low-permeability regions, respectively. The upstream concentration equals the surface water concentration  $c_0$  in case of infiltrating surface water conditions. For exfiltrating groundwater conditions, the upstream concentration equals the concentration of the respective subsurface region.

#### 2.4 Solute transport as a benchmark for microbial transport




In contrast to colloids, which undergo straining in the low-permeability pore space, solute transport occurs equally in the highand low-permeability pore space. Thus, in theory, the transport of conservative solutes could be used as a benchmark for the quantification of size exclusion (i.e., straining) of microbes, so long as the attachment, detachment and inactivation rates are sufficiently well known (Flynn et al., 2006).

Dissolved atmospheric noble gases are conservative tracers for river-aquifer-interactions and provide information on mixing ratios of different water types (Beyerle et al., 1999; Mattle et al., 2001; Schilling et al., 2017; Schilling et al., 2022; Blanc et al., 2024). As noble gases are ubiquitous in the atmosphere, water that is in contact with the atmosphere can be assumed to have an atmospheric noble gas content that is characterized by the air-water exchange equilibrium. Groundwater from different recharge zones can therefore be distinguished by its atmospheric noble gas composition (Kipfer et al., 2002; Aeschbach-Hertig and Solomon, 2013; Cook and Herczeg, 2000). If the recharge zones or compositions of the different mixing components are known, the fractions of the different components in a mixture can be estimated using linear end member mixing analysis, which was demonstrated by Popp et al. (2021). At the wellfield scale, <sup>4</sup>He acts as a conservative tracer for disentangling locally

infiltrated river water and regional groundwater. Once the fractions of locally infiltrated river water and regional groundwater 245 in riverbank filtration wellfields are known, concentrations of the radioactive noble gas 222Rn, which is only produced and present in significant quantities in the subsurface, can be used to estimate the travel times of locally infiltrated river water (Hoehn and Von Gunten, 1989; Cecil and Green, 2000; Vogt et al., 2010; Peel et al., 2022; Peel et al., 2023; Popp et al., 2021). The transport, production and decay of environmental gas tracers in the subsurface as implemented in HGS is described in 250 detail in Delottier et al. (2022). In short, the production of <sup>222</sup>Rn can be simulated by using a zero-order source with partitioning. The production rate per unit pore volume [M L<sup>-3</sup> T<sup>-1</sup>] is the product of the steady-state equilibrium <sup>222</sup>Rn activity, its decay constant, and the porosity. Furthermore, the aqueous/gas phase partitioning coefficient [-] needs to be specified. <sup>4</sup>He production in the subsurface can also be simulated using a zero-order source with partitioning. However, at the wellfield scale, owing to the very low production rate of <sup>4</sup>He, the concentrations of different water components are dominated either by their background concentrations (e.g., that of regional alluvial groundwater entering the wellfield) or the air-water exchange equilibrium and 255 excess air components that characterize freshly recharged water such as locally infiltrated river water (Popp et al., 2021; Schilling et al., 2017). Hence, for simulations of riverbank filtration sites, as long as the background concentrations of the different groundwater components are known, <sup>4</sup>He can be treated like a conservative solute such as a dye, for example.

Moreover, Blanc et al. (2024) showed recently that in settings with only small variations of the natural <sup>4</sup>He concentrations of different water types, the breakthrough of a sufficiently large artificial <sup>4</sup>He injection into the surface water or groundwater can be attributed to the injection experiment and provide an alternative to classical conservative dye tracing.

# 3 Verification



To verify the dual-permeability approach with two-site kinetic deposition mode as implemented in HGS, a synthetic numerical experiment simulating reactive colloid transport through a soil column was performed. The simulation results were compared to an analytical solution for colloid transport in dual permeability media developed by Leij and Bradford (2013). The synthetic experiment was designed after an experiment of Bradford et al. (2009) and consists of a 1 m vertical soil column with a low-and a high-permeability region. Flow is at steady-state and a constant colloid injection pulse is applied to the top of the soil column for a total of 1.2 h. The colloid breakthrough curves simulated by HGS at the bottom of the soil column for the two permeability regions and the retained colloid concentrations on the grain surfaces of the two permeability regions were compared to those of the analytical solution.

#### 3.1 Numerical model set-up

In HGS, the 1D column with a total height of 1 m is discretized with vertical intervals of 0.025 m. A homogeneous and fully saturated soil column with a high- and low-permeability pore space is assumed. The hydraulic conductivities for the high- and low-permeability pore spaces are set to 60 m d<sup>-1</sup> and 40 m d<sup>-1</sup>, respectively. A steady flow field is generated by applying fixed head boundary conditions at the top (inlet,  $h_{in}$ ) and the bottom (outlet,  $h_{out}$ ) of the soil column, such that  $h_{in} - h_{out} = 0.15$  m.

The total porosity is defined as 0.3. Assuming an equal fraction of high- and low-permeability pore space (i.e.,  $w_h = w_l = 0.5$ ), the resulting total fluid flux amounts to 7.5 m d<sup>-1</sup>. In both permeability regions, the longitudinal dispersivity,  $\alpha_l$ , is equal to 0.01 m and tortuosities  $\tau_h$  and  $\tau_l$  are equal to 1.0. The bulk density in both regions is set to 1760 kg m<sup>-3</sup>.

The first-order fluid exchange coefficient  $a_w$  between the two permeability regions is set to 3.0 m<sup>-2</sup>, and the hydraulic conductivity at the interface between the two regions,  $K_a$ , is equal to 0.01 m d<sup>-1</sup>.

The colloid attachment rates  $k_{atth}$  and  $k_{attl}$  are set to 12 d<sup>-1</sup> and 120 d<sup>-1</sup> in the high- and low-permeability region, respectively. In both regions a detachment rate of  $k_{deth} = k_{detl} = 0.024$  d<sup>-1</sup> is assigned, the transfer coefficient for the liquid phases  $\omega_{ex}$  is set to 16 d<sup>-1</sup>, while the coefficient for colloid exchange between the solid phases in the high and low-permeability regions  $k_t$  is set to 0 d<sup>-1</sup>.

The soil column is initially free of colloids. A 1.2 h colloid input pulse using a  $3^{rd}$  type concentration boundary with a colloid inflow concentration of  $1000 \text{ N}_c \text{ m}^{-3}$  is applied and the flow and transport within the column are simulated for 3 h.

In addition to the base-case simulation, the following four scenarios of flow and transport are simulated where (i) the liquid exchange coefficient  $\omega_{ex}$  is increased by a factor of 10, (ii) first-order decay in both liquid regions is assumed to occur with  $\lambda_h = \lambda_l = 12 \text{ d}^{-1}$ , (iii) the detachment rate in both regions ( $k_{deth}$  and  $k_{detl}$ ) is increased by a factor of 100, and (iv) the attachment rate in the high-permeability region  $k_{atth}$  is increased by a factor of 10. All parameters are summarized in Table 1.

# 3.2 Analytical solution for colloid transport in dual-permeability media





The analytical solution derived by Leij and Bradford (2013) is based on the same partial differential equations as outlined for the HGS model implementation (Eq. 10-14) and accounts for advective and dispersive transport in both a high- and a low-permeability region, with reversible kinetic-sorption sites in each region and exchange between the liquid phases. Moreover, the analytical solution includes a second, irreversible retention site in both regions. As outlined by Leij and Bradford (2013), these irreversible retention sites can also be used to account for other first-order sink terms such as, e.g., inactivation. Limitations of the analytical solution for verifying the implementation in HGS are that in the analytical solution the pore space is equally shared by the high- and low-permeability regions ( $w_h = w_l = 0.5$ ), the detachment rates of both regions are equal ( $k_{deth} = k_{detl}$ ), and there is no colloid exchange between the two solid phases ( $k_t = 0$ ).

The same parametrization as used for the numerical experiments was employed for the analytical solution (see Table 1). The spatial and temporal discretization was set to 0.2 m and 0.01 h, respectively. The analytical solution was computed using the original Fortran executable of Leij and Bradford (2013).

Table 1: Simulation base parameters and simulation scenarios as employed for model verification

| parameter                                                 | base case | scenario |       |     |       |
|-----------------------------------------------------------|-----------|----------|-------|-----|-------|
|                                                           |           | i        | ii    | iii | iv    |
| <b>K<sub>h</sub></b> [m d <sup>-1</sup> ]                 | 60        |          |       |     |       |
| $\mathbf{K_l}$ [m d <sup>-1</sup> ]                       | 40        |          |       |     |       |
| $n_{\rm h}=n_{\rm l}$ [-]                                 | 0.3       |          |       |     |       |
| $w_{\rm h} = w_{\rm l}  [-]$                              | 0.5       |          |       |     |       |
| $\alpha_l$ [m]                                            | 0.01      |          |       |     |       |
| $\tau_{ m h} = \tau_{ m l} \; [	ext{-}]$                  | 1.0       |          |       |     |       |
| $ \rho_{bh} = \rho_{bl} \left[ \text{kg m}^{-3} \right] $ | 1760      |          |       |     |       |
| $a_w$ [m <sup>-2</sup> ]                                  | 3.0       |          |       |     |       |
| $K_a$ [m d <sup>-1</sup> ]                                | 0.01      |          |       |     |       |
| $\lambda_{sh} = \lambda_{sl} [d^{-1}]$                    | 0         |          |       |     |       |
| $k_t$ [d <sup>-1</sup> ]                                  | 0         |          |       |     |       |
| $k_{atth}$ [d <sup>-1</sup> ]                             | 12        | 12       | 12    | 12  | 120   |
| $k_{att1}$ [d <sup>-1</sup> ]                             | 120       | 120      | 120   | 120 | 120   |
| $k_{deth} = k_{detl}  [\mathrm{d}^{\text{-}1}]$           | 0.024     | 0.024    | 0.024 | 2.4 | 0.024 |
| $\omega_{ex}$ [d <sup>-1</sup> ]                          | 16        | 160      | 16    | 16  | 16    |
| $\lambda_{\rm h} = \lambda_{\rm l} \ [{ m d}^{-1}]$       | 0         | 0        | 12    | 0   | 0     |

# 3.3 Results



Fig. 1 illustrates the breakthrough of colloids in the liquid phase at the column outlet for the base-case parametrization (Fig. 1a) as well as the final profile of attached colloids at the end of the experiment (Fig. 1b). The concentrations in the low-permeability region as simulated with HGS are depicted in lighter, the concentrations in the high-permeability region in darker shades. The analytical solutions for the low ( $c_1$  and  $s_1$ ) and the combined low+high-permeability ( $c_t$  and  $s_t$ ) spaces are plotted using dashed and solid black lines, respectively. The liquid and the solid concentrations in both regions simulated by HGS are in very good agreement with the analytical solution. At the column inlet, a perfect match was not possible because of the vicinity to the inflow boundary condition and related numerical instabilities in the evaluation of the integrals within the analytical solution.

Figure 1: Comparison of the simulation results in HGS and the analytical solution for a synthetic 1D soil column, showing (a) the breakthrough curve ( $c_t = c_h + c_1$ ) of a 1.2 h colloid injection in the liquid phase at the column outlet and (b) the final retention of the attached colloid concentration  $s_t$  after 3 h, with the two sub-components  $s_h$  and  $s_1$  indicated as well.

The total liquid phase concentrations (Eq. 15) and the corresponding total solid phase concentrations (Eq. 16) for the five scenarios are presented in Fig. 2. Also under varying parametrizations, an excellent fit between the numerical solution of HGS and the analytical solution could be achieved. As mentioned above, minor discrepancies between the analytical and numerical solution at the solid phase column inlet can also be observed for the different scenarios.

Figure 2: (a) Breakthrough curves of total colloid concentration in the liquid phase  $c_t$  at the column outlet and (b) final retention profile of the total attached colloid concentration  $s_t$  for the base-case scenario, as well as scenarios with a (i) a 10 times higher colloid transfer coefficient between the two regions  $\omega_{ex}$ , (ii) first-order decay in both liquid regions ( $\lambda_h = \lambda_l = 12 \text{ d}^{-1}$ ), (iii) a 100 times higher detachment rate in both regions ( $k_{deth}$  and  $k_{detl}$ ), and (iv) a 10 times higher attachment rate in high-permeability region  $k_{atth}$ .




In comparison to the base-case scenario, scenario i (blue markers and line in Fig. 2) has an increased liquid exchange coefficient, and thus, more colloids are transported in the low-permeability pore space, where the attachment rate is higher. This results in a lower breakthrough of the liquid concentration at the column outlet and in an increased final sorbed concentration. In scenario ii (depicted in red in Fig. 2), the total liquid concentration as well as the total sorbed concentration are lower than in the base-case scenario, due to colloid decay in both liquid regions. The increased detachment rate in region 1 (scenario iii, purple markers and line in Fig. 2) affects the liquid breakthrough concentration at later simulation times. The column is initially colloid free, and colloids need to attach first, before the enhanced detachment rate impacts the colloid distribution. Similarly, the solid phase concentration is solely affected at the column inlet, where the colloid pulse arrives first and sufficient time for attachment and enhanced detachment is given. The higher attachment rate in region 1 (scenario iv, depicted in yellow in Fig. 2), results in a more than 10 times lower peak concentration of the breakthrough curve. The total sorbed concentration is significantly higher at the column inlet, however lower at the column outlet, as less colloids reach the column outlet via advective and dispersive transport.

In summary, the dual permeability approach with two-site kinetic deposition mode as implemented in HGS could be verified successfully against the existing analytical solution by Leij and Bradford (2013).

# 4 Illustrative Examples





# 4.1 Streamflow generation by return flow

To illustrate microbial transport in surface water, groundwater, and across the surface water - groundwater interface, an existing benchmark for rainfall-runoff and streamflow generation simulations was used. The benchmark is based on a small-scale sandy catchment located at the Borden research site in Canada that was first studied and simulated by Abdul (1985). The surface topography of this V-shaped catchment consists of gentle hillslopes and a central, runoff collecting depression whose width is about 2 m. Several studies have been based on this benchmark for rainfall-runoff and solute transport simulations with ISSHM (e.g. Jones et al. (2006), Thomas et al. (2016) and De Maet et al. (2015)). Kollet et al. (2017) finally established the setup as a benchmark to compare various ISSHMs in simulating hydrological surface-subsurface interactions and streamflow generation. More recently, Gutiérrez-Jurado et al. (2019) modified the benchmark to systematically investigate the influence of soil types and precipitation intensity on streamflow generation with the HGS model. The simulations presented here are based on the benchmark modified by Gutiérrez-Jurado et al. (2019) where it is assumed that the catchment consists of a sandy gravel soil instead of sand. A 6-hour heavy precipitation event is simulated to illustrate a situation where streamflow generation during rainfall events is predominantly driven by exfiltration of pre-event groundwater and interflow. For the purpose of this study, the model is further extended to include a dual-permeability subsurface. Moreover, a conceptual microbial contamination event in the form of an agricultural manure application shortly before the 6-hour heavy precipitation event is added. To simultaneously track microbial transport, solute transport and streamflow generation mechanisms, advective-dispersive transport of a conservative solute and a hydraulic mixing-cell analysis are also employed.

# 4.1.1 Model set-up

The extent of the model is 80 m, 16 m, and 4.6 m in the x-, y- and z-directions, respectively (Fig. 3). Compared to Gutiérrez-Jurado et al. (2019), the triangular mesh was significantly refined using the mesh-generator AlgoMesh (Hydroalgorithmics Pty Ltd., 2020). The element size ranges from 0.3 m in the runoff collecting depression up to a maximum of 2.0 m at the catchment boundaries. The model is discretized vertically into 15 layers, starting from a layer thickness of 0.1 m at ground surface and gradually increasing with depth to a maximum of 0.5 m at the model bottom. In total, the mesh comprises 64,704 nodes and 119,445 triangular elements.

Manning's roughness coefficient n for the runoff collecting depression and for the surrounding grassy hillslope are equal to 3.5E-06 m<sup>-1/3</sup> d and 3.5E-07 m<sup>-1/3</sup> d, respectively, which are values used by Gutiérrez-Jurado et al. (2019). The sandy gravel has a total porosity of 0.41, van Genuchten parameters  $\alpha$  and  $\beta$  of 16 m<sup>-1</sup> and 1.79, and a residual saturation of 0.045. For the purpose of this study, the subsurface is extended into a dual-permeability medium, consisting in equal proportions of a high-and of a low-permeability region with effective hydraulic conductivities of 171 and 17.1 m d<sup>-1</sup>, respectively. The first-order

fluid exchange coefficient between the two permeability regions is 0.6 m<sup>-2</sup> and the hydraulic conductivity of the interface is 86 m d<sup>-1</sup>.

The simulation is started from an initially dry channel and a uniform initial hydraulic head of 2.78 m in both subsurface permeability regions, resulting in an initial depth to water table of 0.2 m below the channel. At the outlet of the catchment, a critical depth boundary condition is applied in the surface domain. In both permeability subsurface regions, a fluid transfer boundary condition is applied with a reference head of 2.8 m at a distance of 5 m from the downstream model boundary to allow groundwater outflow. Analogously to Gutiérrez-Jurado et al. (2019), a precipitation rate of 0.39 m d<sup>-1</sup> (2<sup>nd</sup> type BC, corresponding to 16 mm h<sup>-1</sup>) is applied to the surface water domain for 6 hours to generate streamflow from exfiltrating groundwater. After 6 hours, the precipitation rate is reduced to a value of 3.9E-7 m d<sup>-1</sup> to maintain a small value of stream discharge until the end of the simulation. The total simulation time is 20 days.

An agricultural application of manure in the upper half of the catchment is mimicked to represent a source of microbial contamination (Fig. 3). The application is assumed to occur immediately before the precipitation event and the manure contains a generic faecal microbial species. In the application area, the initial concentration of the microbial species is 1.0E9 cells m<sup>-3</sup> in the surface domain and at the top of the high- and low-permeability subsurface regions. The attachment and detachment rate coefficients are both equal to 0.0006 d<sup>-1</sup> in the high-permeability region and the attachment and detachment rate coefficients in the low-permeability regions are set to 0.05 d<sup>-1</sup> and 0 d<sup>-1</sup>, respectively. The irreversible attachment in the low-permeability region, resulting from the detachment rate equal to 0 d<sup>-1</sup>, represents straining due to size exclusion of the faecal microbial species. The microbial decay rates in the liquid phase are set to 0.0086 d<sup>-1</sup> in both regions. The decay constants for the solid phases are set to 0.0001 d<sup>-1</sup> and 0.001 d<sup>-1</sup> in the high- and low-permeability region, respectively. The coefficient for microbial transfer between the liquid phases in both regions is defined as 0.01 d<sup>-1</sup> and the coefficient for colloid exchange between the solid phases is set to 0 d<sup>-1</sup>. A first order decay coefficient in the surface domain of 1.0E-5 d<sup>-1</sup> is applied (Brooks and Field, 2016). To provide a comparison for transport of microbes undergoing inactivation and attachment, a conservative solute is also applied simultaneously to the generic faecal microbe during the conceptual manure application phase. Table 2 lists all simulation parameters for the first illustrative example.

To identify and quantify sources of water in the stream discharge, the hydraulic mixing-cell (HMC) method in HGS is used (Partington et al., 2011; Schilling et al., 2017). Three possible water sources for streamflow are considered: direct runoff, return flow (exfiltrating groundwater) from the high-permeability region, and return flow from the low-permeability region. Surface water that infiltrates into the subsurface is immediately marked as groundwater of the respective permeability region, while groundwater that exfiltrates to the surface remains marked as groundwater, allowing to distinguish strictly between direct runoff and water that has infiltrated and travelled through the subsurface. To further enable the separation of return flow into pre-event and event-water (Kirchner, 2003), the groundwater initially present in the system is labelled with a second conservative solute with an initial concentration of 100 kg m<sup>-3</sup>, representing 100% pre-event groundwater. Two virtual

observation points, one in the stream (OPstream, Fig. 3) and one 0.2 m below the stream bed (OPsubsurface, Fig. 3), are located about 20 m upstream of the catchment's outlet to monitor the stream discharge (OPstream) and the contributions of the three water sources as well as the microbial and solute concentrations in the stream (OPstream) and the subsurface (OPsubsurface).

Table 2: Simulation parameters as employed for the illustrative streamflow generation model

| parameter                                          | value   |
|----------------------------------------------------|---------|
| Manning's <i>n</i> (stream) [m <sup>-1/3</sup> d]  | 3.5E-06 |
| Manning's $n$ (grassy land) [m <sup>-1/3</sup> d]  | 3.5E-07 |
| Coupling length $l_{exch}$ [m]                     | 0.002   |
| <b>K</b> <sub>h,aquifer</sub> [m d <sup>-1</sup> ] | 171     |
| <b>K</b> <sub>l,aquifer</sub> [m d <sup>-1</sup> ] | 17.1    |
| $n_{ m h}=n_{ m l}$ [-]                            | 0.41    |
| $w_{\rm h} = w_{\rm l} \ [-]$                      | 0.5     |
| Residual saturation [-]                            | 0.045   |
| Van Genuchten $\alpha$ [m <sup>-1</sup> ]          | 16      |
| Van Genuchten $\beta$ [-]                          | 1.79    |
| $\alpha_l$ [m]                                     | 5.183   |
| $\alpha_t$ [m]                                     | 0.5183  |
| τ [-]                                              | 0.1     |
| $ \rho_{bh} = \rho_{bl} [\text{kg m}^{-3}] $       | 1765    |
| $a_w$ [m <sup>-2</sup> ]                           | 0.6     |
| $K_{a,\text{aquifer}}$ [m d <sup>-1</sup> ]        | 86      |
| $k_{atth}$ [d <sup>-1</sup> ]                      | 0.0006  |
| $k_{att1}$ [d <sup>-1</sup> ]                      | 0.05    |
| $k_{deth}$ [d <sup>-1</sup> ]                      | 0.0006  |
| $k_{det1}$ [d <sup>-1</sup> ]                      | 0       |
| $\omega_{ex}$ [d <sup>-1</sup> ]                   | 0.01    |
| $k_t$ [d <sup>-1</sup> ]                           | 0       |
| $\lambda_{h} = \lambda_{l} [d^{-1}]$               | 0.0086  |
| $\lambda_{sh}$ [d <sup>-1</sup> ]                  | 0.0001  |
| $\lambda_{s1}$ [d <sup>-1</sup> ]                  | 0.001   |
| $\lambda_{\mathrm{o}}  [\mathrm{d}^{\text{-1}}]$   | 1.0E-5  |

# 4.1.2 Results

Fig. 3 shows the simulated concentration of the microbial species in the liquid phase 10 days after a heavy, 6-hour rainfall event. The microbial species gets transported through the subsurface and at the surface along the topographic gradient from the manure application area towards and along the stream channel. In the subsurface, the highest concentrations can be observed directly below the application area due to the water infiltration during and after the heavy rainfall event. Moreover, the microbial species migrates below the stream due to exchange fluxes between the surface and subsurface domains. The plume of microbes below the stream clusters in areas of higher microbial concentrations indicating increased stream water infiltration, and areas of lower microbial concentrations indicating increased groundwater exfiltration. These are typical patterns of hyporheic exchange flow along the stream-aquifer interface, reflecting the bathymetry of the stream bed.

Figure 3: Illustration of the streamflow generation model with microbial transport. The upper 2D surface shows the microbial concentration in the surface domain, while the lower 3D domain shows the microbial concentration in the subsurface. The manure application area in the surface domain is marked by a white dashed line, the observation points in the stream (OPstream) and 0.2 m below the streambed in the subsurface (OPsubsurface) are shown by black cubes. Depicted is the simulated total concentration of the microbial species in the liquid phase 10 days after a heavy, 6-hour rainfall event.

430 Fig. 4 illustrates the temporal evolution of stream discharge, the contributing fractions of the three water types, and the microbial and solute concentrations at the two observation points. The total peak discharge is reached after 0.3 days (Fig. 4a). The fraction of stream water originating from direct runoff peaks shortly after the rainfall, with a short-term contribution

reaching a maximum of about 70%, and it decreases very rapidly to a contribution of less than 0.1% (Fig. 4b). The major contribution to stream discharge is return flow from the high-permeability subsurface region with a value reaching 90% on average for combined pre-event and event water. The contribution of return flow from the low-permeability region for combined pre-event and event water is much lower and its maximum is 10%. As the fraction of direct runoff decreases, the fraction of pre-event groundwater from the high-permeability region increases and stabilizes at about 70%.

Figure 4: (a) Total discharge of the stream and (b) contributions of surface runoff and return flow from the high- and low-permeability subsurface regions during the 20 days simulation period as well as (c) simulated faecal microbial and conservative solute (liquid) concentrations at the observation point in the surface (OPstream,  $c_0$ ) and (d) in the subsurface (OPsubsurface,  $c_t$ ) subdivided into contributions from the high- and low-permeability regions.

With the onset of stream discharge by direct runoff, microbes and solutes rapidly migrate into the stream channel leading to a 445 rapid initial increase of their concentrations in the surface water at OPstream (Fig. 4c). After this rapid initial increase, the rates of increase begin to decline at around 0.6 days. Despite this slowdown, the solute concentration continues to rise and reaches its peak after 3.3 days (Fig. 4c). This subsequent increase in the solute concentration can be attributed to return flow. The peak of the microbial concentration in the stream water is reached 2 days before the peak of the conservative solute 450 concentration (Fig. 4c), which results from the faster apparent transport of microbes in the subsurface due to size exclusion as well as, to a lesser degree, from microbial inactivation in the subsurface and surface domains. At OPsubsurface (Fig. 4d), the first arrival and a minor increase in total liquid solute and microbial concentrations c<sub>t</sub>, barely visible in Fig. 4d, occur shortly after the rapid concentration increase in the river. Concentrations then start to significantly increase after 1.7 days. Return flow dominates stream discharge shortly after the rain event and near the manure application area. Further down the channel 455 hyporheic flow starts to develop, which transports microbes and solutes in and out of the streambed. The microbial concentration at the subsurface observation points peaks 2 days before the conservative solute. This illustrates both the faster apparent transport of colloidal microbes due to size exclusion as well as microbial inactivation.

This illustrative streamflow generation simulation with microbial transport highlights the suitability of the new transport implementation to co-simulate explicit microbial and solute transport with an ISSHM on the surface, in the subsurface and across the surface-subsurface interface - processes and surface-subsurface interactions which cannot be simulated or studied with pure groundwater or pure surface water models in such a rigorous manner.

#### 4.2 Riverbank filtration

As second illustrative example for coupled reactive microbial and solute transport at the wellfield scale, the quasi hypothetical riverbank filtration wellfield model developed by Delottier et al. (2022) was extended to a dual-permeability system and forced by a river flood pulse with temporarily increasing microbial and <sup>4</sup>He concentrations in the river water. The model was built after a real-world alluvial riverbank filtration site in Switzerland characterized by a high-permeability buried paleochannel (Schilling et al., 2022) and set up to illustrate the reactive simulation of environmental gas tracer transport, production and decay (Delottier et al., 2022). It was used to understand and inversely reproduce the impact of preferential flow structures on river-aquifer interactions (Delottier et al., 2023), and to demonstrate modular, multi-variate data assimilation with an ISSHM (Tang et al., 2024). Besides the reactive microbial transport as implemented via the new dual-permeability, two-site kinetic deposition mode, reactive transport of <sup>222</sup>Rn and conservative transport of <sup>4</sup>He were simulated in order to demonstrate the potential of the new transport formulation for the integrated simulation of tracers used for the derivation of mixing ratios and travel times.

# 4.2.1 Model set-up

- The synthetic riverbank filtration wellfield model has a spatial extent of 300x500x30 m in x-, y- and z-direction, respectively (Fig. 5). The alluvial plain and the channel are inclined in y-direction with a slope of 0.003. Along the side of the wellfield (x = 282.9-300 m, y = 0-500 m), a trapezoidal river with a depth of 4 m, a riverbed of 12.9 m width and a riverbank of 4.2 m width with a 0.95 m/m slope is simulated. On the floodplain, at a distance of 65 m to the river (i.e., at x = 235 m), 4 evenly spaced pumping wells are set to extract groundwater from a screened interval ranging from a depth of 9 to 27 m.
- The 2D-triangular mesh was generated using the mesh-generator AlgoMesh (Hydroalgorithmics Pty Ltd., 2020). The triangular element side length is set to increased gradually from 0.75 m near the wells and 1.5 m in the river to 3.5 m throughout the rest of the alluvial plain (see Fig. 5). In total, the model is discretized vertically into 23 layers, with the layer thickness increasing from 0.1 m at the top of the model to 3.0 m at the bottom of the model. The final 3D mesh of the model consists of 622,200 nodes, forming 1,172,678 triangular elements.

Figure 5: Conceptual model of the synthetic riverbank filtration site model. The upper 2D surface shows the initial water depth in the surface domain. The lower 3D plot shows the subsurface domain with 4 abstraction wells that are evenly spaced in the alluvial plain at a distance of 65 m from the river. The observation point is at x = 275 m and y = 250 m. The initial hydraulic head distribution before the flood wave is shown and it indicates infiltration from the river and drawdowns around the pumping wells.

The surface roughness is parameterized using a Manning's roughness coefficient *n* of 8.1E-06 m<sup>-1/3</sup> d for the floodplain and 1.91E-07 m<sup>-1/3</sup> d for the river. Streamflow is conceptualized by applying a specified flux boundary condition (2<sup>nd</sup> type BC) at the upstream river nodes and a critical depth BC at the downstream end of the river. The initial river discharge of 0.3 m<sup>3</sup> s<sup>-1</sup> for the model spin-up represents low flow conditions.

The dual node approach is chosen to couple the surface and subsurface domains with a coupling length of 0.007 m and 0.001 m in the floodplain and the river, respectively.






The subsurface is split into two homogeneous zones, an alluvial aquifer and a 0.5 m thick riverbed zone underneath the riverbed and riverbank. The entire subsurface is furthermore extended to a dual-permeability domain, assuming 50% high and 50% low-permeability pore space. The effective hydraulic conductivities are set to 166 m d<sup>-1</sup> and 1.66 m d<sup>-1</sup> for the high and low-permeability zones of the aquifer, and 5 m d<sup>-1</sup> and 0.05 m d<sup>-1</sup> for the high and low-permeability zones of the riverbed, respectively. The first-order fluid exchange coefficient between the high- and low-permeability regions is set to 0.6 m<sup>-2</sup> for both the riverbed and the aquifer. The hydraulic conductivity of the interface is set to 150 m d<sup>-1</sup> in the aquifer and 1.5 m d<sup>-1</sup> in the riverbed. In the entire subsurface, the porosity is set to 0.15. For all zones and regions, the residual saturation is set to 0.05 and the van Genuchten parameters  $\alpha$  and  $\beta$  to 3.4 m<sup>-1</sup> and 1.71, representing typical values for alluvial sandy gravel deposits (Dann et al., 2009; Schilling et al., 2021).

Regional groundwater flow is accounted for by a fixed head BC (1<sup>st</sup> type BC) in the high- and low-permeability regions at the upstream and downstream end of the aquifer. The regional hydraulic gradient is set to equal the slope of the aquifer, and the water table is at the elevation of the riverbed which is 4 m below the surface of the wellfield. Each of the 4 pumping wells abstracts water with a constant rate of 805 m<sup>3</sup> d<sup>-1</sup>.

To assess mixing ratios, <sup>4</sup>He is defined as a conservative solute, which is a valid assumption for travel times associated with this wellfield configuration. The concentration at the upstream boundary of the aquifer is set to 5.81E-8 cm<sup>3</sup><sub>STP</sub> g<sub>H2O</sub><sup>-1</sup>, which corresponds to the <sup>4</sup>He-concentration that was observed in the alluvial aquifer of the Swiss pre-Alps after which the synthetic model was designed (Schilling et al., 2017). At the nodes of the upstream river boundary condition, the concentration is defined as 4.3E-8 cm<sup>3</sup><sub>STP</sub> g<sub>H2O</sub><sup>-1</sup>, which corresponds to an air-water-equilibrium at a water temperature of 6.4°C and air pressure of 923 hPa - average winter conditions at the Swiss reference study site. As a proxy for groundwater age, the production and decay of the natural radioactive noble gas isotope <sup>222</sup>Rn is considered and simulated following Delottier et al. (2022). The upstream boundary condition is set to the production-decay equilibrium activity of 15 Bq L<sup>-1</sup>, a typical secular equilibrium for sandy gravel aquifers (Peel et al., 2022; Popp et al., 2021). Due to the absence of <sup>222</sup>Rn in the atmosphere, the activity at the surface nodes upstream river boundary condition is set to 0 Bq L<sup>-3</sup>. The first-order decay constant of <sup>222</sup>Rn is set to its decay rate of 0.181 d<sup>-1</sup>. The <sup>222</sup>Rn secular equilibrium, the <sup>222</sup>Rn decay rate, and the porosity of the subsurface result in a production rate of 0.40725 Bq L<sup>-1</sup> d<sup>-1</sup> for both regions. The water/gas partitioning coefficient is set to 0.35.

Table 3: Simulation parameters as employed for the riverbank filtration model

| parameter                                                       | value    |  |
|-----------------------------------------------------------------|----------|--|
| Manning's n (river) [m <sup>-1/3</sup> d]                       | 8.1E-06  |  |
| Manning's $n$ (floodplain) [m <sup>-1/3</sup> d]                | 1.91E-07 |  |
| Coupling length $l_{exch,river}$ [m]                            | 0.001    |  |
| Coupling length $l_{exch, floodplain}$ [m]                      | 0.007    |  |
| <b>K</b> <sub>h,aquifer</sub> [m d <sup>-1</sup> ]              | 166      |  |
| <b>K</b> <sub>l,aquifer</sub> [m d <sup>-1</sup> ]              | 5        |  |
| $\mathbf{K}_{\mathbf{h}, \text{riverbed}}$ [m d <sup>-1</sup> ] | 16.6     |  |
| <b>K</b> <sub>l,riverbed</sub> [m d <sup>-1</sup> ]             | 0.5      |  |
| $n_{\rm h}=n_{ m l}$ [-]                                        | 0.15     |  |
| $w_{\rm h} = w_{\rm l} \ [\text{-}]$                            | 0.5      |  |
| Residual saturation [-]                                         | 0.05     |  |
| Van Genuchten $\alpha$ [m <sup>-1</sup> ]                       | 3.4      |  |
| Van Genuchten $\beta$ [-]                                       | 1.71     |  |
| $\alpha_l$ [m]                                                  | 5.183    |  |
| $\alpha_t$ [m]                                                  | 0.5183   |  |
| τ [-]                                                           | 0.1      |  |
| $ \rho_{bh} = \rho_{bl} \text{ [kg m}^{-3}\text{]} $            | 1765     |  |
| $a_w$ [m <sup>-2</sup> ]                                        | 0.6      |  |
| $K_{a,aquifer}$ [m d <sup>-1</sup> ]                            | 150      |  |
| $K_{a,\text{riverbed}}$ [m d <sup>-1</sup> ]                    | 1.5      |  |
| $k_{atth}$ [d <sup>-1</sup> ]                                   | 0.06     |  |
| $k_{att1}$ [d-1]                                                | 0.005    |  |
| $k_{deth}$ [d <sup>-1</sup> ]                                   | 0.06     |  |
| $k_{detl}$ [d <sup>-1</sup> ]                                   | 5E-5     |  |
| $\omega_{ex}$ [d <sup>-1</sup> ]                                | 0.01     |  |
| $k_t$ [d <sup>-1</sup> ]                                        | 0        |  |
| $\lambda_h = \lambda_l \ [d^{-1}]$                              | 0        |  |
| $\lambda_{sh} = \lambda_{sl} [d^{-1}]$                          | 0.001    |  |

Microbes are simulated as total cell count (TCC), representing all microbial cells in the system. The total cell concentration flowing into the aquifer is defined as 30,000 cells mL<sup>-1</sup>, the microbial cell concentration at the surface to 300,000 cells mL<sup>-1</sup>. These concentrations are in the typical range of cell concentrations present in Swiss surface water and groundwater (Kötzsch and Sinreich, 2014). The parametrization of the reactive microbial transport processes is based on literature values for prokaryotes and bacteriophages that were originally estimated by fitting transport models with two first-order kinetic sites to

observations of experiments at the well-field scale (Sasidharan et al., 2018; Schijven and Šimůnek, 2002; Hornstra et al., 2018; Kvitsand et al., 2015). The first-order rates in the high-permeability region are set to 0.06 d<sup>-1</sup> for attachment and 0.06 d<sup>-1</sup> for detachment. The attachment and detachment coefficients in the low-permeability regions are set to 0.005 d<sup>-1</sup> and 5E-5 d<sup>-1</sup>, respectively. The rates in the low-velocity regions are by one order of magnitude smaller than the ones in the high-permeability region. Thereby, the concept of fast and reversible attachment in the high-permeability region and slow and (almost) irreversible attachment in the low-permeability region (Bradford et al., 2009) is incorporated. No microbial decay is simulated in the liquid phases. The decay constants for the solid phases are set to 0.001 d<sup>-1</sup> ensuring that the fraction of microbes removed from the liquid phase due to straining are eliminated irreversibly. The coefficient for microbial transfer between the liquid phases in both regions is defined as 0.01 d<sup>-1</sup> and the coefficient for colloid exchange between the solid phases is set to 0 d<sup>-1</sup>. Table 3 lists all simulation parameters of the illustrative example.

A model spin-up of 10,000 days with constant BC was performed. The relatively long model spin-up was necessary to reach steady-state for both liquid and solid phase concentrations. The initial hydraulic head and concentration distributions are illustrated in Fig. 5 and Fig. 6 and represent steady-state and low flow conditions.

Over a simulation period of 20 days, a discharge event, peaking at 1.75 days with a maximum inflow rate of 12.3 m<sup>3</sup> s<sup>-1</sup>, is simulated. The time varying flux BC is illustrated in Fig. 7. Microbial concentrations in river water generally rise during flood events (Ferguson et al., 2003). Therefore, the TCC concentration of the inflowing river water increases proportionally to the discharge up to the double of the initial TCC concentration (600,000 cells mL<sup>-1</sup> during the maximum inflow rate at simulation time 1.75 days). As a comparison for the reactive microbial transport, an equivalent peak of the conservative tracer <sup>4</sup>He is simulated (8.6E-8 cm<sup>3</sup><sub>STP</sub> g<sub>H2O</sub><sup>-1</sup> during the discharge peak), mimicking an artificial gas injection tracer test, as for example conducted by Blanc et al. (2024).

# 4.2.2 Results







The initial total flux concentrations of the situation prior to the arrival of the flood are illustrated in Fig. 6. The distribution of the conservative mixing tracer <sup>4</sup>He indicates that between river and abstraction wells, the groundwater consists of 100% river water that infiltrated within the model domain. On the side of the wells facing away from the river, however, regional groundwater dominates. The <sup>222</sup>Rn activity concentration increases gradually from the river to the abstraction wells, showing increasing residence times of the freshly infiltrated river water in the aquifer. At the abstraction wells, the <sup>222</sup>Rn activity is already close to the steady-state equilibrium. The highest microbial cell counts in the liquid phase as well as attached to the solid phase occur within the 0.5 m thick riverbed. With increasing travel distance in the subsurface, the liquid phase cell concentration decreases significantly due to straining, attachment and inactivation of the microbes. At the abstraction wells the total cell count is 20% lower than in the river water. Due to the decreasing TCC in the liquid phase and inactivation of microbes on the grain surfaces, also the total solid phase concentration of the microbes decreases with distance to the river.

Figure 6: Simulated steady-state concentrations of <sup>4</sup>He, <sup>222</sup>Rn, TCC (liquid phase) and TCC (solid phase) in the subsurface before the arrival of the flood wave.

The simulated total concentration timeseries in the liquid phase during the flood event at the near-river observation point are depicted in Fig. 7. For comparison, the TCC, <sup>4</sup>He and <sup>222</sup>Rn concentrations are normalized by their initial concentrations. During the flood event, the TCC and <sup>4</sup>He concentrations in the inflowing river water increased proportionally to the discharge up to twice the inflow concentrations, causing a clear breakthrough at the piezometer of both TCC and <sup>4</sup>He. The maximum concentration occurs 4.65 days (TCC) or 4.95 days (<sup>4</sup>He) after the discharge peak in the river. Due to size exclusion, microbial transport predominantly occurs in the high-permeability pore space, resulting in the microbial peak arriving 0.3 days before the one of the (conservatively treated) solute tracer <sup>4</sup>He. Due to attachment and inactivation of the microbes, the concentration peak of the normalized TCC is smaller than the one of the conservative <sup>4</sup>He. The longer tailing of the microbial peak compared to <sup>4</sup>He is caused by initial attachment followed by a slow detachment (i.e., release) of microbes from the solid phase back into the liquid phase. The temporary decrease in the <sup>222</sup>Rn activity indicates that, in reaction to the flood, the groundwater at the observation point became younger. As opposed to the microbial and <sup>4</sup>He breakthrough curves, which reflect reactive advective-dispersive solute and colloid transport from the river to the piezometer, the change in the <sup>222</sup>Rn activity at the piezometer can

be attributed to a shifting flow pattern. The maximum variation in the <sup>222</sup>Rn activity therefore occurs nearly immediately after the increase of discharge in the river, hence 4 days before the breakthrough of the microbes and <sup>4</sup>He.



In summary, the illustrative simulations demonstrate the suitability of the two-site kinetic deposition, dual-permeability flow and transport formulation in HGS to co-simulate reactive microbial and solute transport on the well-field scale. The simulated concentrations show faster transport of microbes due to size exclusion compared to the slower bulk transport of solutes like <sup>4</sup>He, which is a known phenomenon. The combined impact of attachment, detachment and inactivation processes on the microbial concentration could, in turn, be pointed out by comparing the shape of the microbial breakthrough curve to the one of the conservatively transported <sup>4</sup>He. The parallel simulation of <sup>222</sup>Rn production and decay in the dual-permeability subsurface moreover allowed to include a natural tracer for the characterization of residence times of freshly infiltrated river water in bank filtration contexts. Ultimately, the illustrative example demonstrates the potential of the newly developed tool to simulate the simultaneous transport of multiple different environmental tracers and microbes in an integrated surface-subsurface hydrological model.

Figure 7: River discharge during the 20-days flood event and simulated concentrations of TCC (liquid), <sup>4</sup>He and <sup>222</sup>Rn (all normalized by the initial concentrations  $c_0$ ) in the high and low-permeability regions at the observation point x = 275 m, y = 250 m.

# 5 Discussion







Having implemented and verified the dual-permeability approach with two-site kinetic deposition mode, HydroGeoSphere is the first ISSHM that is able to simulate water flow, heat as well as reactive solute and colloid transport in parallel and throughout the surface and subsurface domains.

Transforming the subsurface into a dual-permeability system doubles the number of subsurface nodes, which makes the simulation computationally more expensive. The increased computational demand might currently still limit the application of the tool to wellfield, reach and small catchment scale studies. However, computational power increases continuously and rapidly, and cloud computing, which is commercially now widely accessible and highly scalable, reduces simulation times significantly (Kurtz et al., 2017).

The dual-permeability, two-site kinetic deposition implementation is highly flexible for reactive solute and microbial transport, but it also requires defining additional flow and transport parameters. Such a complex model inevitably requires multiple and diverse types of observations, most importantly multi-tracer data, in order to allow for a robust calibration of the different parameters, which is necessary for any surface water-groundwater or larger scale groundwater model (Schilling et al., 2019). In contrast to the original development of the dual-permeability model describing anomalous flow and transport in fractured rocks with distinct head and flow differences between the two domains (Gerke and Van Genuchten, 1993), the application of the dual-permeability approach for microbial transport in a porous media subsurface shows minimal head differences between the two regions. Consequently, concentration data needs to inform about the fractions of the two regions and the ratio of the two hydraulic conductivities. This means that concentration data of a conservative solutes and colloids, namely microbes, undergoing size exclusion in the low-permeable, small-pore region are required. The specific combination of environmental gas and microbial data, as for example presented in the illustrative riverbank filtration example, is a particularly promising data set to calibrate such models. Environmental (noble) gases can be analysed continuously in situ (Brennwald et al., 2016) and they have been shown to improve ISSHM calibration significantly (Schilling et al., 2022; Schilling et al., 2017). In addition, online flow cytometry (FCM) now allows to monitor microbial dynamics in riverbank filtration settings on site, continuously, and in near-real time (Besmer et al., 2016). Beyond direct cell counting, microbial clusters can be distinguished, such as high (HNA) and low (LNA) nucleic acid content microbes, which are often referred to as larger and smaller procaryotes and are a typically observed cytometric pattern in aquatic systems (Wang et al., 2009). As the developed tool enables to cosimulate the transport of environmental gases and reactive microbes, the ISSHM can be calibrated against both types of tracer data simultaneously, and the identifiability of model parameters as well as the data worth of each tracer type towards reducing the uncertainty of model parameters and model predictions can be evaluated. This combination will be of huge relevance for the identification of the transport mechanisms of pathogens in riverbank filtration settings, and the subsurface in general, and enable more targeted field campaigns for the robust delineation of WHPA.

The microbial diversity in the aquatic environment is enormous. For simplicity and the purpose of illustration of the capability of the developed model, we represented the entire microbial community in the illustrative riverbank filtration example by the

total cell count. While for the study of microbial processes, this reduction in complexity is certainly an oversimplification, the colloid transport formulation in HGS enables multispecies transport. This capability could thus be harnessed to subdivide the microbial community into the typical flow cytometric derived HNA and LNA groups or to simulate the transport behaviour of specific waterborne pathogens, for example. The downside of increasing the number of microbial groups is the increasing computational requirements. Hence, the type of microbes or microbial groups to be simulated should be chosen wisely. On the other hand, the more the different microbial clusters or groups are discriminated, the more information about attachment, detachment and inactivation rates may be derived from a well calibrated model, and thus, the filtration efficiency of the subsurface for specific microbial groups be quantified.

The development of this integrated hydrological-microbial simulation tool is an important step towards improving our understanding of microbial transport processes in riverbank filtration settings and in quantifying them. Beyond, it can become a valuable tool for drinking water management e.g., to design WHPA and evaluate the risk of drinking water contamination by pathogens also under environmental stress factors such as extreme weather situations, flood events or revitalization of rivers.

#### **6 Conclusions**





The dual-permeability model with two-site kinetic deposition mode for the simulation of reactive microbial transport in river-groundwater systems was implemented in the ISSHM HydroGeoSphere (available in revision 2699 and onwards). The model formulation allows to simulate preferential microbial transport in the high-permeability subsurface region and slower solute bulk transport in both the high- and low-permeability region in parallel, under consideration of both surface and subsurface fluxes in fully integrated manner. The implementation was verified successfully against an analytical solution for modified colloid transport in dual-permeability media. As demonstrated by the illustrative examples, the method is well suited to study microbial transport across the river-aquifer interface and to co-simulate reactive microbial transport in riverbank filtration settings combined with environmental gas tracer transport. With the development of this tool, it is now possible to calibrate an ISSHM against online microbial and environmental gas monitoring data, and by this, to quantify transport and removal processes of microbes in the subsurface, which is crucial to improve drinking water management at riverbank filtration sites and protect drinking water against contamination of pathogens. The new model might prove to become an important decision support tool for real-time operational wellfield management.

Code and Data availability. All model input files required to run the verification and illustrative examples are available for download from HydroShare under https://doi.org/10.4211/hs.84911fe3496f44eebbce512a6e8d0db3. To run the models, the

HydroGeoSphere version rev. 2699 (or onwards) as well as a valid license are necessary. A trial version of HydroGeoSphere can be obtained from aquanty.com.

Author contributions. FC: Conceptualization, Methodology, Validation, Investigation, Visualization, Writing – Original Draft, Review & Editing. RT: Software, Validation, Writing – Review & Editing. OS: Conceptualization, Methodology, Validation, Investigation, Writing – Review & Editing, Supervision, Project administration, Funding acquisition.

**Competing interests.** There are no conflict of interests.

Acknowledgements. Friederike Currle gratefully acknowledges the funding of the Sentinel North grant (Internship Scholarship Program for International Students). Additional funding has been received through the SNSF-JSPS Strategic Japanese-Swiss Science and Technology Programme (SJSSTP) grant no. 214048.

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
