# Peer review of "Explicit simulation of microbial transport with a dual-permeability, two-site kinetic deposition formulation using the integrated surface-subsurface hydrological model HydroGeoSphere"

_EGUsphere, 2025_

## Author Comment (AC1)

**Responses to comments of anonymous reviewer #1 for manuscript 10.5194/egusphere-2025-372**

This manuscript presents the implementation of an existing dual-permeability approach to address transport of (microbial) colloids in the proprietary software HydroGeoSphere that can simulate saturated-unsaturated flow and transport coupled to surface flow and transport. The code is validated against an existing analytical solution for 1-D dual-permeability transport with linear exchange kinetics between the two domains and linear kinetic sorption in both of them. The final demonstration is a virtual test case mimicking bank filtration, in which colloidal transport is simulated together with 4He and radon as natural tracers. There is no comparison to experimental or field data.

We would like to thank the reviewer for the time and effort invested in the review of our manuscript and note that no scientific concerns on the algorithm and code implementation have been raised. Below we indicate the feedback to the specific suggestions for revision of our manuscript.

As the manuscript's major statement is that the authors have transferred an existing model formulation to an existing software package I see this as a technical note rather than a research article.

The reviewer is correct, this is a technical note. However, we would like to point out that we didn't simply transfer an existing formulation to an existing software package but combined an existing formulation for subsurface microbial transport based on dual-permeability, two-site kinetic deposition with a decay term to account for physical straining and/or death of microbes and integrated this to a surface-subsurface reactive solute and colloid transport solution. This is thus the first integration of multiple transport model formulations for solutes and colloids in a fully integrated surface-subsurface flow and transport model.

The 1-D example is not particularly exciting from a process-insight view because the influence of different parameters on multi-permeability models has been discussed before. I have remarks on the 3-D application further down.

As the section title clearly indicates, the purpose of the 1-D example was to validate the numerical implementation of the code on a well-described, existing analytical benchmark. It is a well-founded and important common practice to compare a new feature to another model (numerical or analytical) in order to ensure that the feature has been correctly coded.

With respect to presenting the model extension of HydroGeoSphere, the text is written in a slightly odd way. It is not clear what was actually developed within the study, and what has already existed. I checked the HydroGeoSphere documentation, which includes colloid transport in dual-permeability models. But that might only mean that the documentation has already included the results of the present study.

The new development, namely the integration of an existing dual-permeability, two-site kinetic deposition approach for microbial transport in a fully integrated surface-subsurface flow and transport model, was clearly outlined in the description of the study aims on lines 102-109 of the original manuscript. What was missing from this statement was the fact that we also added a decay term to the formulation. However, this information was given on lines 175-177. We will add this information also to the aim statement in the introduction.

With respect to the release notes: One of the authors, René Therrien, is the origin developer of HydroGeoSphere. As he co-developed the module used here, it has always been documented in the version updates for HydroGeoSphere starting from the time of the first code implementations. The first time the new implementation was announced in the HydroGeoSphere release notes was in September 2023 (revision 2582) and the setup and use of it was included in the corresponding HydroGeoSphere manuals. Subsequent revisions fixed bugs (revisions 2596, 2633) or added additional features for microbial colloid transport (revision 2699) that were coming up during the development of this study. This will be clarified in the revised manuscript.

The only wording that directly implies model extension is in lines 175-177, where the inclusion of a first-order decay term is mentioned. Please clarify that the implementation of the Bradford et al. (2009, doi: 10.1029/2008WR007096) formulation is really specific to this study. Otherwise I am even more puzzled what the message of the manuscript is.

We will clarify the implementation aspects in the aims section of the manuscript, as outlined above. In addition, we would like to note that our illustrative example is also the first demonstration of explicit co-simulation of microbial transport with reactive solute transport (of noble gas radioisotopes) in a fully integrated surface-subsurface hydrological model. The new aspects thus go beyond the pure addition of an algorithm to an existing software. This information is also already provided in the aims statement of the original manuscript.

Let's come to the conceptual assumptions of the Bradford et al. (2009) formulation. Dual-permeability flow and transport was introduced by Barenblatt et al. (1960, doi: 10.1016/0021-8928(60)90107-6) to address preferential flow in fractured media, and has been used to parameterize effects on heterogeneity on (flow and) solute transport, characterized by strong anomalous behavior. The concept has been presented using different names (e.g., multi-domain model, mobile-mobile model). In contrast to dual-porosity (mobile-immobile, transient storage) models, it requires solving two coupled flow problems, posing big difficulties in unique calibration if the flow behavior is quite normal. Bradford et al. (2009) reinterpreted the conceptual model to facilitate that colloids break through earlier than solutes. The latter effect is caused by size exclusion and by the fact that colloids cannot experience the velocity within pores at distances to solids smaller than half their diameter. If the colloids have the same net electrical charge as the grain surface they experience an even smaller portion of the intra-pore velocity distribution because they are repelled from the no-slip boundary. Other model formulations for colloid transport, involving reversible attachment-detachment and irreversible straining, cannot reproduce a first breakthrough before that of solutes. Bradford et al. (2009) claimed that their formulation also addresses straining, but that is not really true. In both domains, the model assumes kinetic first-order attachment and detachment. It is trivial to derive equilibrium sorption coefficients from the ratio of the detachment and attachment rate coefficients. Choosing a low detachment coefficient in the low-permeability domain still implies reversible sorption. However, straining is irreversible: particles get stuck in pore throats and never ever become mobile again. This implies that the formulation of Bradford et al. (2009) leads to very long tailing of colloid breakthrough curves with a complete recovery at infinite times, whereas models that include an elimination mechanism according to standard colloid filtration theory will have an incomplete recovery (i.e. the zeroth moment of the transfer function between the in- and output concentration in 1-D transport is smaller than unity). If you really want straining, you either choose a detachment coefficient in the less-mobile domain of zero, or you introduce a first-order elimination term. The authors have such a term, but they relate it other

mechanisms than straining because they uncritically adopt the erroneous perspective of Bradford et al. (2009) that kinetic reversible mass transfer could parameterize (irreversible) straining.

The reviewer is correct, the Bradford et al. 2009 implementation does not allow for reversible attachment-detachment in both high- and low-permeability domains alongside irreversible straining without completely eliminating reversible attachment-detachment in the low permeability zone, which doesn't do justice to the actual processes. To address this, we indeed included an elimination term on top of the implementation of Bradford et al., 2009. The reviewer correctly points out that we missed to fully describe this in the original manuscript. We will add a proper description of this in the revised manuscript.

The reviewer is also correct that in the illustrative case we simplified the simulation by superimposing reversible attachment-detachment with irreversible straining in the low-permeability domain. However, we applied a first-order elimination term on the attached microbes in the low-permeability domain, which ensures that a fraction of attached microbes do not detach at infinite times, giving an incomplete recovery. As described above, we will point out more clearly the different processes in the revised manuscript.

Would there be a much simpler way of achieving an earlier breakthrough of colloids than of solutes without introduction of a second permeability? That is indeed possible. All you need is a retardation factor for the colloids smaller than one, and they are faster than the solutes. You would still need an irreversible straining term and (at least one) kinetic, reversible attachment-detachment term, but the model would be much simpler because you could skip coming up with two spatial permeability distributions and exchange terms of the fluids between the two domains (and the flow problem would have half the number of unknowns). Given the fact that the authors don't want to use the dual-permeability formulation for its original purpose (addressing anomalous flow and transport in highly heterogeneous formations), and that they don't have the data to inform such a model for its original purpose, I cannot recommend the conceptual approach chosen by the authors. It is simply an overkill, particularly in 3-D settings.

Indeed, simpler and oversimplified implementations are always possible. However, the goal of this implementation was to apply scientific rigor and to improve system understanding, particularly the differences of solute vs colloid/microbial transport on reach- to catchment-scales. With the new implementation in HydroGeoSphere, we for the first time provide a tool that can do so for fully coupled surface-subsurface hydrological systems. Where tracer time series, for example of $^{222}$Rn, $^{4}$He and/or microbes, exist, one can now use our implementation to understand system parameters via inverse modelling. This will lead to a significant decrease in the ill-position of the problem of inversely identifying hydraulic properties relevant for solute vs pathogen transport in alluvial aquifer systems. In central Europe, these systems are the most widely used systems for drinking water production. Having thus a tool that finally allows solving or at least further constraining the parameters that are relevant for the transport of solute contaminants and colloidal pathogens is a very important breakthrough. We see this as a great scientific tool and overkill not as a scientific criterion.

The authors promise a reactive microbial transport model in the title. This is slightly misleading. The only "reaction" term is a first-order elimination term. Many microbes of interest undergo complex dynamics due to growth, dormancy and reactivation, change in mobility corresponding to their physiological state or abundance. In the model of the authors

the microbes must be introduced via the inflow and can only adsorb or vanish besides transport. They are essentially treated like dead particles. I can fully understand that the authors don't intend to elaborate on microbial dynamics, but then they should be careful in selling they model as "reactive microbial transport model".

We agree, the term reactive transport model is slightly misleading and does not do justice to the complex reactions microbes can engage in and undergo. We remove the term "reactive" from the title in the revised version of the manuscript, but would like to note that the model nevertheless allows for reactive transport simulations of multiple solute and colloidal species.

As mentioned above, I am not too impressed by the 1-D tests. They mainly reproduce the work of Bradford et al. (2009), who at least had a comparison to real measurements, and of Leij and Bradford (2013). It is of course important that a code is tested against analytical solutions for model validation, but it is not a scientifically particularly exciting exercise.

Indeed, the point is to compare the code against a previously well-described and scientifically proven analytical benchmark model. The scientific criterion is simply that the numerical implementation must reproduce the analytical solutions. This is demonstrated in a very clear manner in the manuscript, which is potentially also why it appears slightly unimpressive. There is nothing overly exciting to discuss about it, as the reproduction is nearly perfect. And this is true over many different parametrizations.

The 3-D demonstration is supposed to show that the model works field-similar settings. It is not well suited to convince the reader that an integrated surface-subsurface hydrological model is needed. The test would equally well work with a pure groundwater model forced by the boundary condition at the river. In the setup, like in many bank filtration applications, the river is not really affected by groundwater flow and transport (and every HydroGeoSphere user knows that running the model as pure porous-medium model makes life much easier). A more interested application would show real feedbacks between the surface and subsurface domains, e.g., when considering transport of pathogens in a meandering stream with intensive hyporheic exchange and bank storage. That is obviously not the final application that the authors have in mind, but I would claim that the 3-D test problem could be simulated with loose coupling of the surface and subsurface domains (or even just predefining river-stage and concentration fluctuations as boundary condition).

Why integrated models are needed for certain situations is demonstrated widely in literature, for example in the expansive reviews by Paniconi and Putti (10.1002/2015WR017780) or Simmons et al. (10.1016/j.jhydrol.2019.124309). Our illustrative model is not meant to show that an integrated simulator is absolutely needed, it is meant to demonstrate how a runoff event with increased microbial load in the surface water results in an increased microbial load also in the groundwater, not by assigning a boundary condition that mimics surface water on the top of the subsurface domain, but by assigning a high microbial load directly to the river water entering the surface domain. The illustrative example thus shows how HydroGeoSphere with the new implementation can handle dual-permeability, two-site kinetic deposition based microbial transport alongside reactive transport of solutes in a fully coupled river-aquifer system. Many different other examples could have been chosen but we indeed intended to show the capability on one of the intended uses which is the most relevant system for drinking water production in Europe, namely bank filtration wellfields.

While in our eyes the illustrative model shows exactly what it needs to show, in the revised manuscript we are happy to add another illustrative model based on an implementation of the Borden benchmark model by Gutierrez-Jurado et al. (10.1029/2019WR025041), which shows rainfall-runoff generation and, in contrast to the riverbank filtration benchmark, not only produces infiltration but also generates return flow from the subsurface to the surface.

In the 3-D test case, the authors first simulate steady-state concentrations, involving an input of (microbial) colloids from the river, $^4$He as a natural tracer for the mixing of old groundwater with river water, and $^{222}$Rn as age tracer. They then simulate the response to a river-stage fluctuation. Proportional to the river discharge they increase the concentration of the (microbial) colloids an of $^4$He in the river (mimicking an artificial tracer with concentration is proportional to that of the microbes). The former makes sense, whereas I am not happy about the later because $^4$He is supposed to be an indicator of mixing of old groundwater and river infiltrate, and with the pulse in the river the boundary between the two water bodies moves. While this boundary may be far away from the observation points, it is not particularly smart to create two causes of $^4$He changes. It might have been better to add a real artificial tracer to separate the signals.

We simulated the excitingly new scenario where $^4$He instead of a dye is injected into stream water for artificial tracing purposes. The illustrative example is tailored after the very recent experiments by Blanc et al (10.1016/j.watres.2024.121375) and Brennwald et al. (10.3389/frwa.2022.925294). This was clearly stated on lines 406-408 of the original manuscript. However, we do agree completely that this is so new and exciting that it may be confusing and agree that the $^4$He pulse overlaps with natural $^4$He signals. To make it clearer why we simulated a $^4$He pulse and address the reviewer's concern, we will revise the description of the illustrative example and highlight the artificial $^4$He tracing more clearly.

As designed, the colloids break slightly earlier through than the solute tracer. The effect is not super big (4.5% earlier peak time) and well within the uncertainty of travel-time estimates in real-world studies on outlying well-head protection zones. With a longitudinal dispersivity of more than 5m (and additional dispersion caused by mobile-mobile transport), the peaks are so broad that the difference in the breakthrough curves are not particularly obvious by eye sight. The much more interesting signal is that of the radon. Here, the authors get a much earlier breakthrough. They attribute this to radon following the pressure wave (lines 433-434), but that makes physically no sense. What I believe is that the river-stage fluctuation shifts the flow pattern and by that the age distribution. Honestly, I find this phenomenon more interesting than the micobial-transport study as you see a real signal.

We agree with the reviewer that the behavior of the $^{222}$Rn signal is very interesting, and we will certainly use the new tool to study such parallel dynamics more thoroughly in the near future. As the purpose of this illustrative example was simply to show the implementation for microbial transport in HydroGeoSphere and the possibility to simulate in parallel microbial transport and reactive solute transport, an in-depth analysis and discussion of the reasons for the changes in the $^{222}$Rn breakthrough profile are outside of the scope of this study. We will add a note that the changes in the breakthrough curve could also be related to shifting flow patterns.

The results section ends with a summary/conclusion, followed by a discussion section, and then final conclusions. That's a little bit odd, particularly since the actual conclusions are quite shallow.

Since this is a technical note in which we first present the integrated and developed code, subsequently provide a verification of the numerical model against an analytical solution, and finally illustrate the capabilities of the model on one example, a classic paper structure is not ideal. We therefore decided to provide results of the verification first, as these can be considered the most important when it comes to numerical model implementation. We finish this with a short discussion of these results. The illustrative model and its results are subsequently presented and discussed separately. Finally, we provide an overall discussion section to discuss more generally the results and implications of using an integrated model for explicit co-simulation of microbial and reactive solute transport, and put this in context to the availability and need for data. Conclusions are kept short on purpose, as this is a technical note.

In summary, I have expressed my doubts that the dual-permeability model is the best choice for transport of microbial colloids. I am convinced that you can achieve the same results computationally much cheaper. I believe that the 1-D model has too much weight given that it includes nothing new. The 3-D application does not need an integrated surface-subsurface model and does not underscore that the chosen model formulation is really needed. If there were real data that can only be interpreted with the model, the authors would have a much stronger point. This manuscript needs severe revisions to make it a significant contribution.

We agree that multiple ways of simulating microbial transport, some simpler and more efficient, some more complex and more computationally demanding, exist. We wanted to strike a balance between computational efficiency and complexity, and selected among the available tools the most reasonable combination to allow for explicit co-simulation of reactive solute and microbial transport on reach- to catchment-scales with an integrated surface-subsurface hydrological model. As this is a technical note, the 1-D verification is a must, and substantial weight should be put on the verification of the numerical implementation - whether it is exciting or not is irrelevant. The 3-D illustrative case of a bank filtration wellfield with explicit surface water-groundwater interaction and transport is exactly where such a model will provide new insights via co-simulation and inversion of solute and microbial transport. We agree that an additional illustrative model that includes infiltration and return flow will be a great addition to the manuscript and will provide this in the revised manuscript. Presenting and reproducing real data is however outside of the scope of this study. There are no studies which so far even provide reasonable time series of such tracers in such a context for a prolonged period of time (even only multiple days to weeks). We are in the process of producing and publishing such datasets, and we are sure that the reviewer agrees that a technical note on a model extension is not the right place to publish such data. We will do so in a separate research article and finally demonstrate the combination and implementation of such data in the integrated simulation of a real-world bank filtration wellfield and inverse identification of the dominant transport parameters and mechanisms.

---

## Author Comment (AC2)

**Responses to comments of anonymous reviewer #2 for manuscript 10.5194/egusphere-2025-372**

This manuscript outlines the inclusion of dual-permeability, two-site kinetic deposition formula for microbial transport in HydroGeoSphere (HGS). I believe this topic is relevant and of interest to the readers of EGUsphere. The manuscript is generally well-written, with the inclusions of expected sections outlining the model development, validation and illustrative application, all written in a clear manner.

We would like to thank the reviewer for the time and effort invested in reviewing our manuscript. We would like to point out that the manuscript is a technical note and not a research paper - a mistake that happened during the submission process.

As presented, my primary concern relates to the novelty and contributions of this work. How does this differentiate from other subsurface reactive transport models available – why should someone choose to use HGS over these models? I understand that the primary feature of including these equations in HGS is the inclusion with an integrated hydrologic model as opposed to solely a subsurface model, but no part of this manuscript uses or highlights the benefit of an integrated approach over subsurface-only. The authors outline the equations that govern surface flow and transport, but the verification and illustrative application do not seem to utilize the surface domain at all. Perhaps it is simulated, but the results are not presented or discussed. I think this is critical to the contributions of this work – what does this newly developed feature provide that was otherwise lacking? Perhaps a comparison between groundwater-only simulations and the integrated approaches can help demonstrate the benefit of including these equations with HGS as opposed to solely a subsurface approach.

Given this concern is critical to the contributions and novelty of this research, I feel the authors need to make significant revisions before this manuscript can be considered for publication.

We agree that the novelty lies in the combination of dual-permeability, two-site kinetic deposition based microbial transport with an integrated surface-subsurface model that is capable of explicitly simulating reactive solute transport, for example of noble gas radioisotopes. The capability of the integrated model is demonstrated explicitly in the 3-D illustrative model, where both increased microbial and Helium concentrations in the surface water (here the river) are transported into and through the subsurface during a peak flow event. Indeed this could also be oversimplified into a pure groundwater model, as many users of single domain models would argue most situations that concern groundwater systems would allow. Our illustrative example nevertheless demonstrates the capability of the new implementation. While it isn't the purpose or aim of our manuscript to highlight the importance of using integrated surface-subsurface hydrological models in general - this has been demonstrated and reviewed extensively already elsewhere, e.g. by Paniconi and Putti (10.1002/2015WR017780) or Simmons et al. (10.1016/j.jhydrol.2019.124309) - we agree that an additional illustrative case can highlight the potential of the new implementation even better. In the revised manuscript, we will thus include an additional illustrative model based on an implementation of the Borden benchmark model by Gutierrez-Jurado et al. (10.1029/2019WR025041), which shows rainfall-runoff generation and also generates return flow from the subsurface to the surface. The model will thus illustrate not only microbial transport from the surface into the subsurface, but at the same time the inverse, which should

satisfy the desire of seeing a yet more complex situation that benefits from the application a fully integrated simulator.

---

## Author Response (AR1)

This document provides a point-by-point reply to the comments of the anonymous reviewers #1 and #2 regarding the revision of manuscript 10.5194/egusphere-2025-372 and outlines all relevant changes made in the revised version.

**1. Reply to comments of anonymous reviewer #1**

This manuscript presents the implementation of an existing dual-permeability approach to address transport of (microbial) colloids in the proprietary software HydroGeoSphere that can simulate saturated-unsaturated flow and transport coupled to surface flow and transport. The code is validated against an existing analytical solution for 1-D dual-permeability transport with linear exchange kinetics between the two domains and linear kinetic sorption in both of them. The final demonstration is a virtual test case mimicking bank filtration, in which colloidal transport is simulated together with 4He and radon as natural tracers. There is no comparison to experimental or field data. As the manuscript's major statement is that the authors have transferred an existing model formulation to an existing software package I see this as a technical note rather than a research article.

We would like to thank the reviewer for the time and effort invested in the review of our manuscript and happily note that no scientific concerns on the algorithm and code implementation have been raised. The reviewer is correct, this is a technical note, and we have communicated this to the editors.

We would like to point out, however, that we didn't simply transfer an existing formulation to an existing software package. For the present manuscript, we combined an existing formulation for subsurface microbial transport based on dual-permeability, two-site kinetic deposition with a decay term to account for physical straining and/or death of microbes, and we integrated this to a fully coupled surface-subsurface reactive solute and colloid transport solution. This is thus the first integration of multiple transport model formulations for solutes and colloids in a fully integrated surface-subsurface flow and transport model.

The 1-D example is not particularly exciting from a process-insight view because the influence of different parameters on multi-permeability models has been discussed before. I have remarks on the 3-D application further down.

The purpose of the 1-D example was to validate the numerical implementation of the code on a well-described, existing analytical benchmark. This is a common and necessary practice and allows ensuring that a new feature has been correctly coded. Hence, the verification example has to remain an integral part of the manuscript even if it doesn't provide new process insights.

With respect to presenting the model extension of HydroGeoSphere, the text is written in a slightly odd way. It is not clear what was actually developed within the study, and what has already existed. I checked the HydroGeoSphere documentation, which includes colloid transport in dual-permeability models. But that might only mean that the documentation has already included the results of the present study.

The new development, namely the integration of an existing dual-permeability, two-site kinetic deposition approach for microbial transport in a fully integrated surface-subsurface flow and transport model, has been outlined in the description of the study aims on lines 102-109 of the original manuscript. What was missing from this statement was the fact that we

also added a decay term to the microbial transport formulation - an information which had only been given on lines 175-177 of the original manuscript. We thank the reviewer for pointing out that this was unclear. To make the novelty in this technical note more comprehensive to the reader, we added the information that we extended the existing formulation by Bradford et al. (2009) with first-order decay terms in the aim statement of the introduction along with a note about the release versions of HydroGeoSphere (lines 109-113 of the revised manuscript).

With respect to the release notes: One of the authors, René Therrien, is the original developer of HydroGeoSphere. As he is still involved with the company that now produces HydroGeoSphere and co-developed the module used here, the new development has continuously been documented in the version updates for HydroGeoSphere. The first time the new implementation was announced in the HydroGeoSphere release notes was in September 2023 (revision 2582) and the setup and use of it was included in the corresponding HydroGeoSphere manuals. In subsequent revisions we fixed bugs (revisions 2596, 2633) or added additional features for microbial colloid transport (revision 2699) that were coming up during the development of this study. This is now clarified in the revised manuscript.

The only wording that directly implies model extension is in lines 175-177, where the inclusion of a first-order decay term is mentioned. Please clarify that the implementation of the Bradford et al. (2009, doi: 10.1029/2008WR007096) formulation is really specific to this study. Otherwise I am even more puzzled what the message of the manuscript is.

In addition to those lines, the new aspects of our manuscript have now been more clearly stated in the aims section of the manuscript, as explained in the previous comment. We would like to note that our illustrative riverbank filtration example is also the first demonstration of explicit co-simulation of microbial transport with reactive solute transport (of noble gas radioisotopes) in a fully integrated surface-subsurface hydrological model. The new aspects thus go beyond the pure addition of an algorithm to an existing software. This information has been provided in the aims statement of the original manuscript (lines 106-109).

Let's come to the conceptual assumptions of the Bradford et al. (2009) formulation. Dualpermeability flow and transport was introduced by Barenblatt et al. (1960, doi: 10.1016/0021-8928(60)90107-6) to address preferential flow in fractured media, and has been used to parameterize effects on heterogeneity on (flow and) solute transport, characterized by strong anomalous behavior. The concept has been presented using different names (e.g., multidomain model, mobile-mobile model). In contrast to dual-porosity (mobile-immobile, transient storage) models, it requires solving two coupled flow problems, posing big difficulties in unique calibration if the flow behavior is quite normal. Bradford et al. (2009) reinterpreted the conceptual model to facilitate that colloids break through earlier than solutes. The latter effect is caused by size exclusion and by the fact that colloids cannot experience the velocity within pores at distances to solids smaller than half their diameter. If the colloids have the same net electrical charge as the grain surface they experience an even smaller portion of the intra-pore velocity distribution because they are repelled from the noslip boundary. Other model formulations for colloid transport, involving reversible attachment-detachment and irreversible straining, cannot reproduce a first breakthrough before that of solutes. Bradford et al. (2009) claimed that their formulation also addresses straining, but that is not really true. In both domains, the model assumes kinetic first-order attachment and detachment. It is trivial to derive equilibrium sorption coefficients from the ratio of the detachment and attachment rate coefficients. Choosing a low detachment

coefficient in the low-permeability domain still implies reversible sorption. However, straining is irreversible: particles get stuck in pore throats and never ever become mobile again. This implies that the formulation of Bradford et al. (2009) leads to very long tailing of colloid breakthrough curves with a complete recovery at infinite times, whereas models that include an elimination mechanism according to standard colloid filtration theory will have an incomplete recovery (i.e. the zeroth moment of the transfer function between the in- and output concentration in 1-D transport is smaller than unity). If you really want straining, you either choose a detachment coefficient in the less-mobile domain of zero, or you introduce a first-order elimination term. The authors have such a term, but they relate it other mechanisms than straining because they uncritically adopt the erroneous perspective of Bradford et al. (2009) that kinetic reversible mass transfer could parameterize (irreversible) straining.

The reviewer is correct, the Bradford et al. 2009 implementation does not allow for reversible attachment-detachment in both high- and low-permeability domains alongside irreversible straining without completely eliminating reversible attachment-detachment in the low permeability zone, which doesn't do justice to the actual processes. To address this, we indeed included an elimination term on top of the implementation of Bradford et al., 2009. The reviewer correctly points out that we missed to fully describe this in the original manuscript.

The reviewer is also correct that in the illustrative case we simplified the simulation by superimposing reversible attachment-detachment with irreversible straining in the low-permeability domain. However, we applied a first-order elimination term on the attached microbes in the low-permeability domain, which ensures that a fraction of attached microbes does not detach at infinite times, giving an incomplete recovery.

To address the reviewer's concerns, we added a detailed description of the two approaches to incorporate irreversible straining (irreversible attachment in the low-permeability region or first-order sink terms) in the method section (lines 192-197). Furthermore, we added comments in the illustrative examples on the approach of straining we used (lines 379-381 and lines 522-523, respectively).

Would there be a much simpler way of achieving an earlier breakthrough of colloids than of solutes without introduction of a second permeability? That is indeed possible. All you need is a retardation factor for the colloids smaller than one, and they are faster than the solutes. You would still need an irreversible straining term and (at least one) kinetic, reversible attachment-detachment term, but the model would be much simpler because you could skip coming up with two spatial permeability distributions and exchange terms of the fluids between the two domains (and the flow problem would have half the number of unknowns). Given the fact that the authors don't want to use the dual-permeability formulation for its original purpose (addressing anomalous flow and transport in highly heterogeneous formations), and that they don't have the data to inform such a model for its original purpose, I cannot recommend the conceptual approach chosen by the authors. It is simply an overkill, particularly in 3-D settings.

Indeed, simpler and oversimplified implementations are always possible. However, the goal of this implementation was to apply scientific rigor and to improve system understanding, particularly the differences of solute vs colloid/microbial transport on reach- to catchment-scales. With the new implementation in HydroGeoSphere, we for the first time provide a tool

that can do so for fully coupled surface-subsurface hydrological systems. Where tracer time series, for example of 222Rn, 4He and/or microbes, exist, one can now use our implementation to understand system parameters via inverse modelling. This will lead to a significant decrease in the ill-position of the problem of inversely identifying hydraulic properties relevant for solute vs pathogen transport in alluvial aquifer systems. In central Europe, these systems are the most widely used systems for drinking water production. Having thus a tool that finally allows solving or at least further constraining the parameters that are relevant for the transport of solute contaminants and colloidal pathogens is a very important breakthrough. We see this as a great scientific tool and overkill not as a scientific criterion.

The authors promise a reactive microbial transport model in the title. This is slightly misleading. The only "reaction" term is a first-order elimination term. Many microbes of interest undergo complex dynamics due to growth, dormancy and reactivation, change in mobility corresponding to their physiological state or abundance. In the model of the authors the microbes must be introduced via the inflow and can only adsorb or vanish besides transport. They are essentially treated like dead particles. I can fully understand that the authors don't intend to elaborate on microbial dynamics, but then they should be careful in selling they model as "reactive microbial transport model".

We agree that the term reactive transport model does not do justice to the complex reactions microbes can engage in and undergo. We removed the term "reactive" from the title in the revised version of the manuscript but would like to note that the model nevertheless allows for reactive transport simulations of multiple solute and colloidal species.

As mentioned above, I am not too impressed by the 1-D tests. They mainly reproduce the work of Bradford et al. (2009), who at least had a comparison to real measurements, and of Leij and Bradford (2013). It is of course important that a code is tested against analytical solutions for model validation, but it is not a scientifically particularly exciting exercise.

As mentioned above, it is an important common practice to validate new numerical code implementations against existing analytical models. The comparison thus remains an integral part of our manuscript.

The 3-D demonstration is supposed to show that the model works field-similar settings. It is not well suited to convince the reader that an integrated surface-subsurface hydrological model is needed. The test would equally well work with a pure groundwater model forced by the boundary condition at the river. In the setup, like in many bank filtration applications, the river is not really affected by groundwater flow and transport (and every HydroGeoSphere user knows that running the model as pure porous-medium model makes life much easier). A more interested application would show real feedbacks between the surface and subsurface domains, e.g., when considering transport of pathogens in a meandering stream with intensive hyporheic exchange and bank storage. That is obviously not the final application that the authors have in mind, but I would claim that the 3-D test problem could be simulated with loose coupling of the surface and subsurface domains (or even just predefining river-stage and concentration fluctuations as boundary condition).

Why integrated models are needed for certain situations is demonstrated widely in literature, for example in the expansive reviews by Paniconi and Putti, 2015 (doi: 10.1002/2015WR017780) or Simmons et al., 2020 (doi: 10.1016/j.jhydrol.2019.124309). Our illustrative riverbank filtration model is not meant to show that an integrated simulator is

absolutely needed, it is meant to demonstrate how a runoff event with increased microbial load in the surface water results in an increased microbial load also in the groundwater, not by assigning a boundary condition that mimics surface water on the top of the subsurface domain, but by assigning a high microbial load directly to the river water entering the surface domain. The illustrative example thus shows how HydroGeoSphere with the new implementation can handle dual-permeability, two-site kinetic deposition based microbial transport alongside reactive transport of solutes in a fully coupled river-aquifer system. Many different other examples could have been chosen but we indeed intended to show the capability on one of the intended uses which is the most relevant system for drinking water production in Europe, namely bank filtration wellfields.

While in our eyes the illustrative model shows exactly what it needs to show, for the revised manuscript we followed the reviewer's desire to see an even more integrated illustrative model application of our code and have added an illustrative case based on a benchmark model for physically-based rainfall-runoff and streamflow generation simulations. The model is an implementation of the Borden benchmark model as set up by Gutierrez-Jurado et al., 2019 (doi: 10.1029/2019WR025041). In contrast to the riverbank filtration benchmark, the Borden site benchmark model not only produces infiltration but also generates return flow from the subsurface to the surface and hyporheic exchange. The model and model results are presented in Section 4.1 of the revised manuscript.

In the 3-D test case, the authors first simulate steady-state concentrations, involving an input of (microbial) colloids from the river, 4He as a natural tracer for the mixing of old groundwater with river water, and 222Rn as age tracer. They then simulate the response to a river-stage fluctuation. Proportional to the river discharge they increase the concentration of the (microbial) colloids an of 4He in the river (mimicking an artificial tracer with concentration is proportional to that of the microbes). The former makes sense, whereas I am not happy about the later because 4He is supposed to be an indicator of mixing of old groundwater and river infiltrate, and with the pulse in the river the boundary between the two water bodies moves. While this boundary may be far away from the observation points, it is not particularly smart to create two causes of 4He changes. It might have been better to add a real artificial tracer to separate the signals.

We simulated the excitingly new scenario where 4He instead of a dye is injected into stream water for artificial tracing purposes. The illustrative example is tailored after the very recent experiments by Blanc et al., 2024 (doi: 10.1016/j.watres.2024.121375) and Brennwald et al., 2022 (doi: 10.3389/frwa.2022.925294). This was clearly stated on lines 406-408 of the original manuscript. To better clarify the use of natural 4He as a commonly used mixing tracer and the new method behind the artificial 4He injection, we distinguished between the two approaches at the end of the introduction (lines 121-123of the revised manuscript). Secondly, we added a more detailed explanation of the artificial 4He injection in the method section (lines 246-248 of the revised manuscript).

As designed, the colloids break slightly earlier through than the solute tracer. The effect is not super big (4.5% earlier peak time) and well within the uncertainty of travel-time estimates in real-world studies on outlying well-head protection zones. With a longitudinal dispersivity of more than 5m (and additional dispersion caused by mobile-mobile transport), the peaks are so broad that the difference in the breakthrough curves are not particularly obvious by eye sight. The much more interesting signal is that of the radon. Here, the authors get a much earlier breakthrough. They attribute this to radon following the pressure wave (lines 433-434), but

that makes physically no sense. What I believe is that the river-stage fluctuation shifts the flow pattern and by that the age distribution. Honestly, I find this phenomenon more interesting than the micobial-transport study as you see a real signal.

We agree with the reviewer that the behavior of the 222Rn signal is very interesting, and we will certainly use the new tool to study such parallel dynamics more thoroughly in the near future. As the purpose of this illustrative example was simply to show the implementation for microbial transport in HydroGeoSphere and the possibility to simulate in parallel microbial transport and reactive solute transport, an in-depth analysis and discussion of the reasons for the changes in the 222Rn breakthrough profile are outside of the scope of this study. We added a note that the changes in the 222Rn breakthrough profile can be related to shifting flow patterns on lines 560-562 of the revised manuscript.

The results section ends with a summary/conclusion, followed by a discussion section, and then final conclusions. That's a little bit odd, particularly since the actual conclusions are quite shallow.

Since this is a technical note in which we first present the integrated and developed code, subsequently provide a verification of the numerical model against an analytical solution, and finally illustrate the capabilities of the model on one example, a classic paper structure is not ideal. We therefore decided to provide results of the verification first, as these can be considered the most important when it comes to numerical model implementation. We finish this with a short discussion of these results. The illustrative model and its results are subsequently presented and discussed separately. Finally, we provide an overall discussion section to discuss more generally the results and implications of using an integrated model for explicit co-simulation of microbial and reactive solute transport, and put this in context to the availability and need for data. Conclusions are kept short on purpose, as this is a technical note.

In summary, I have expressed my doubts that the dual-permeability model is the best choice for transport of microbial colloids. I am convinced that you can achieve the same results computationally much cheaper. I believe that the 1-D model has too much weight given that it includes nothing new. The 3-D application does not need an integrated surface-subsurface model and does not underscore that the chosen model formulation is really needed. If there were real data that can only be interpreted with the model, the authors would have a much stronger point. This manuscript needs severe revisions to make it a significant contribution.

We agree that multiple ways of simulating microbial transport, some simpler and more efficient, some more complex and more computationally demanding, exist. We wanted to strike a balance between computational efficiency and complexity, and selected among the available tools the most reasonable combination to allow for explicit co-simulation of reactive solute and microbial transport on reach- to catchment-scales with an integrated surface-subsurface hydrological model. As this is a technical note, the 1-D verification is a must, and substantial weight should be put on the verification of the numerical implementation - whether it is exciting or not is irrelevant. The 3-D illustrative case of a bank filtration wellfield with explicit surface water-groundwater interaction and transport is exactly where such a model will provide new insights via co-simulation and inversion of solute and microbial transport. We agree that an additional illustrative model that includes infiltration and return flow is a great addition to the manuscript and provided this in the revised manuscript (Section 4.1).

Presenting and reproducing real data is however outside of the scope of this study. There are no studies which so far even provide reasonable time series of such tracers in such a context for a prolonged period of time (even only multiple days to weeks). We are in the process of producing and publishing such datasets, and we are sure that the reviewer agrees that a technical note on a model extension is not the right place to publish such data. We will do so in a separate research article and finally demonstrate the combination and implementation of such data in the integrated simulation of a real-world bank filtration wellfield and inverse identification of the dominant transport parameters and mechanisms.

In addition, to increase readability and clarity overall, we have also revised the color-scheme of Figure 2 and its references in the plain text (lines 321 and 323) to show consistent coloration throughout the entire manuscript.

**2. Reply to comments of anonymous reviewer #2**

This manuscript outlines the inclusion of dual-permeability, two-site kinetic deposition formula for microbial transport in HydroGeoSphere (HGS). I believe this topic is relevant and of interest to the readers of EGUsphere. The manuscript is generally well-written, with the inclusions of expected sections outlining the model development, validation and illustrative application, all written in a clear manner.

We would like to thank the reviewer for the time and effort invested in reviewing our manuscript. We would like to point out that the manuscript is a technical note and not a research paper - a mistake that happened during the submission process.

As presented, my primary concern relates to the novelty and contributions of this work. How does this differentiate from other subsurface reactive transport models available – why should someone choose to use HGS over these models? I understand that the primary feature of including these equations in HGS is the inclusion with an integrated hydrologic model as opposed to solely a subsurface model, but no part of this manuscript uses or highlights the benefit of an integrated approach over subsurface-only. The authors outline the equations that govern surface flow and transport, but the verification and illustrative application do not seem to utilize the surface domain at all. Perhaps it is simulated, but the results are not presented or discussed. I think this is critical to the contributions of this work – what does this newly developed feature provide that was otherwise lacking? Perhaps a comparison between groundwater-only simulations and the integrated approaches can help demonstrate the benefit of including these equations with HGS as opposed to solely a subsurface approach.

Given this concern is critical to the contributions and novelty of this research, I feel the authors need to make significant revisions before this manuscript can be considered for publication.

We agree that the novelty lies in the combination of dual-permeability, two-site kinetic deposition based microbial transport with an integrated surface-subsurface model that is capable of explicitly simulating reactive solute transport, for example of noble gas

radioisotopes. The capability of the integrated model is demonstrated explicitly in the 3-D illustrative model, where both increased microbial and Helium concentrations in the surface water (here the river) are transported into and through the subsurface during a peak flow event. Indeed this could also be oversimplified into a pure groundwater model, as many users of single domain models would argue most situations that concern groundwater systems would allow. Our illustrative example nevertheless demonstrates the capability of the new implementation. While it isn't the purpose or aim of our manuscript to highlight the importance of using integrated surface-subsurface hydrological models in general - this has been demonstrated and reviewed extensively already elsewhere, e.g. by Paniconi and Putti, 2015 (doi: 10.1002/2015WR017780) or Simmons et al., 2020 (doi: 10.1016/j.jhydrol.2019.124309) - we agree that an additional illustrative case highlights the potential of the new implementation even better. In the revised manuscript, we therefore included an additional illustrative model based on an integrated surface-subsurface hydrological benchmark model of the Borden site as set up by Gutierrez-Jurado et al., 2019 (doi: 10.1029/2019WR025041), which shows rainfall-runoff generation and also generates return flow from the subsurface to the surface (Section 4.1 in the revised manuscript). The new illustrative case highlights not only microbial transport from the surface into the subsurface, but at the same time the inverse, namely return flow and hyporheic exchange.

In addition, to increase readability and clarity overall, we have also revised the color-scheme of Figure 2 and its references in the plain text (lines 321 and 323) to show consistent coloration throughout the entire manuscript.

---

## Author Response (AR2)

This document provides a point-by-point reply to the comments of the anonymous reviewers #2 and #3 regarding the revision of manuscript 10.5194/egusphere-2025-372, Revision 2, and outlines all relevant changes made in the newly revised version.

**1. Reply to comments of anonymous reviewer #2**

This is a very interesting paper, and highly relevant to HESS's diverse readership. It presents a very complex model that generalizes dual permeability flow in a river bed to bacterial residence in the fluid and solid bulk. It is well written, clearly presented, and overall of high quality. The main aspects I found that need improvement are: 1. The model presentation, specifically clearly presenting how each specific aspect is incorporated within the equations, yet not only in the parameterization part, but also on how each process is depicted as a functional form within the equation. 2. What is the added value of the bacterial parametrization implementation, making the observed results differ from other models like colloidal transport or sorption, as the results themselves are not different in a noticeable way? I'm sure there are differences, but the paper will benefit from outlining them explicitly.

We thank the reviewer for the positive feedback and the comments, which helped to further improve our manuscript. Based on the line numbering, it appears as if the reviewer's comments refer to the original manuscript submitted before revisions. We can happily state that several comments were already addressed during the previous round of revisions. Please find our detailed answers to all comments below.

The specific comments are as follows:

The abstract is well-written and provides motivation for the study, as well as its aims. I would suggest adding the implications of this study, specifically, what are "meaningful WHPA delineation and risk assessments even under extreme hydrological situations such as flood events."

Lines 35-45 provide an excellent motivation.

As suggested by the reviewer, we clarified the implication of this study by adding the microbial contamination context to the well head protection area (WHPA) delineation and risk assessment under extreme hydrological situations discussion. This can be found on lines 28-29 of the newly revised manuscript, where we now state:

"It enables meaningful WHPA delineation and risk assessments of riverbank filtration sites with respect to microbial contamination even under extreme hydrological and microbial stress situations such as flood events"

Line 48: Does diffusion really play a role for bacteria? Maybe for 0.1 μm, but it will not have any effect on the process of early breakthrough

It was shown previously that particularly for motile bacteria the diffusion coefficient plays a dominant role especially for low-velocity groundwater flow (e.g. Bradford et al., 2014).

Line 62: "secondary energy minimum" is not defined.

Thank you for finding this missing definition. We added a definition of the secondary energy minimum to line 67-69 of the newly revised manuscript. It now reads:

"Colloid attachment under unfavorable conditions is either fast and reversible for colloids that are retained in the secondary energy minimum (i.e. shallow energy minimum for weak colloid

attraction), or slow and irreversible for colloids that overcome the repulsive energy barrier and reach the primary energy minimum (Tufenkji, 2007)"

General remark for the introduction: The introduction is well-written and clear; however, there are not many references to actual experimental results, nor to the possible parameters that may govern these experiments on bacterial/viral migration in porous media. Are there no relevant experiments in the literature?

There are relevant experiments, but to keep it short and concise, in the introduction, we aimed to give only a brief summary of the history of microbial transport, which started several decades ago, and included some recent studies that are directly relevant for the conceptualization of microbial or viral transport as implemented in our model. Instead of mentioning them in the introduction, we refer to recent experimental studies that estimated microbial transport parameters in Section 4.2.1, where we introduce and describe microbial transport in the illustrative riverbank filtration example (lines 5214-527 of the previously revised manuscript).

Line 118: "by a modified form of Richards' equation." What are the modifications? I know that a detailed model will be presented later, but a brief preview of where we differ from the Richards equation can make the following part of the model easier to understand.

The original Richards' equation was developed for flow in unsaturated soils. The modified form was developed to apply it to variably-saturated soils, where some portions of the soil can be fully-saturated, for example below the water table where water storage is a function of the specific storage of the soil, which does not appear in the original Richards' equation. The modification is the addition of specific storage to the storage term in Richards' equation. Some earlier studies that used the modified form include Cooley (1971), Neuman (1973), Huyakorn et al. (1984), as well as Therrien and Sudicky (1996) who present the model on which HydroGeoSphere was initially based. Since the primary focus of this study is the implementation of microbial transport rather than the specifics of variably-saturated flow, we considered that discussing modifications to the Richards' equation is beyond the scope of this work, but we included a note regarding the modification of the specific storage in the saturated zone along with a reference to the complete formulations (to Therrien and Sudicky, 1996) on line 136-137 of the newly revised manuscript. For information, the references for Cooley (1971), Neuman (1973), Huyakorn et al. (1984) are listed below but they have not been added to the paper.

Cooley, R.L., 1971. A finite difference method for unsteady flow in variably saturated porous media: Application to a single pumping well. Water Resour. Res.. 7(6): 1607-1625.

Huyakorn. P.S., Thomas. S.D. and Thompson. B.M., 1984. Techniques for making finite elements competitive in modeling flow in variably saturated porous media. Water Resour. Res., 20(8): 1099-1115.

Neuman. S.P., 1973. Saturated-unsaturated seepage by finite elements. Proc. Am. Soc. Civ. Eng., J. Hydraul. Div., 99(HYl2): 2233-2251.

Line 130: "k" is generally noting the permeability or intrinsic permeability. Consider using a different notation for the relative permeability or adding a subscript, as it is more related to occupancy of the fluid within the porous structure and less to the actual permeability, which is included in the hydraulic conductivity. I believe that in any case, it should be written as k\_rl.

We realized there was a typo in the main text. We already used "k\_r" for the relative permeability in the equation, which is consistent with the suggestions provided by the reviewer. Thank you for pointing this out. We corrected the variable notation in the main text (line 150 of the newly revised manuscript).

Also, is the code implementing the model available in a repository or on GitHub? In the "Code and Data availability" section, the links lead to the input files, but I couldn't access the trial version of HydroGeoSphere even after the extensive verification stage.

All the equations that were considered and combined are given and discussed in the manuscript. Hence, the transport formulations can be readily implemented in any modeling software. As HGS is a proprietary numerical modelling software that belongs to the company Aquanty, Inc., we are not able to provide the specific code implementation in HGS. However, for reproduction of our verification or illustrative examples as well as other test models, a free trial version of HGS can be obtained from aquanty.com directly. The model input files for these examples as well as a statement on the availability of the test license is what is provided in the "Code and Data availability" section.

Equations 10-14: This section as a whole, and the equations part specifically, are not presented in a way that allows the reader to understand the model. Each specific equation represents a different aspect: equations 10 and 11 refer to the high and low concentration transport, respectively, and equations 12 and 13 refer to the density. Although the sentence preceding this one states this aspect, it requires the reader to understand it without specific guidance. I recommend dividing the equations according to their representation (separating them into 10 and 11, 12 and 13, and 14 alone) and providing a detailed explanation for each part, marking the role of each aspect in the equation, rather than just referring to each parameter within it. As an example, take the important aspect of the "microbial inactivation in the liquid and solid phases" marked by  $\lambda$ , the first-order sink term. This is the extension of the authors' from a colloidal model to a bacterial model; hence, it is a seminal aspect in the study. Why not explain why the structure of  $\lambda sl\rho blsl$  marks the decay form in the solid? When the model is complex, an effort should be made to guide the reader on the role of each aspect in the model.

Thank you for the comment. We agree and split and rearranged the equations and the definition of the variables in the main text to make it more reader-friendly (lines 191-202 in the newly revised manuscript).

Equation 17 is a tensorial ADE with an exchange and sink term, and since this is a known form, it's easier to follow. However, I believe that equation 18, which depicts the exchange between the surface to the subsurface, heavily relies on the previous section. Yet this is not clearly explained.

To improve clarity, we added a short note mentioning that we refer to the high- and low-permeability regions that were introduced in the previous section (line 222-223 in the newly revised manuscript).

Section 3.1. A paragraph should be added explaining how well the synthetic experiment matches natural conditions and which natural conditions it replicates.

The aim of the verification example was to show that the implemented feature is correctly coded. For this purpose, we designed a model which is comparable with an existing analytical

solution. This analytical solution was developed based on a colloidal laboratory column transport experiment by Bradford et al. (2009) as outlined on lines 252-253 of the previously revised manuscript. The scenario can be considered as a typical situation in a riverbed, which is fully saturated, but in the absence of complex biochemical reactions that typically happened due to the strong biogeochemical activity of the hyporheic zone. Since the analytical equation itself was developed based on a laboratory column experiment, we refrained from over interpreting this aspect by providing such explanations and believe it is better to instead discuss the transferability to real world scenarios in the illustrative example sections.

Line 279. Correct the term: "low+high-permeability"

This term refers to "low-permeability + high-permeability" and is thus a correct abbreviation.

Line 282. Can you provide the integral evaluation for the analytical solution in an appendix?

As noted in lines 290–291 of the previously revised manuscript, we employed the original published Fortran executable by Leij and Bradford (2013), which includes a detailed description of the underlying mathematics and numerical integration. Since we used this existing tool without modifying its internal numerical procedures, for further technical details, readers are referred to the original publication.

Figure 2. Looking at the BTC and colloidal adsorption to the column, the results are strikingly similar to the solute sorption-desorption process in soil. Can the authors comment on what the observable difference is stemming from the colloidal nature of this simulation, specifically referring to the various scenarios in Table 1, and even more specifically, to scenario 2, where the decay rate is introduced?

Furthermore, what is the reasoning for providing a decay rate of ~2 hours for a pulse of 2 hours? Wouldn't it make a competing aspect where, prior to attachment, there will be a decay in considerable values? And isn't that the explanation for the rapid decay with the depth, namely, that the exchange can only occur at the upper layers, prior to the decay occurring over time?

Can the authors comment on the model's sensitivity to various parameters? Which is more dominant: the exchange rate, conductivity difference for the dual permeability, or perhaps the reactive components of the attachment\detachment?

As mentioned before, the purpose of the verification example was exclusively to show that the new numerical feature is correctly coded and able to reproduce an analytical solution. The parameter variations shown in Fig. 2 were selected to provide a clear and intuitive visual indication of the correct implementation across a broad range of parameter values. We believe that a detailed analysis of individual parameter combinations would not substantially contribute to the core focus of the paper, which is the simulation of microbial transport at the wellfield scale. For this aspect, we subsequently provide two very illustrative scenarios which we also discuss at length. To avoid repeating previous work, we refer readers to Bradford et al. (2009) and other studies comparing colloid and solute transport for a more thorough discussion of parameter sensitivity in the dual-permeability framework.

Line 399: What were the steady-state criteria?

The entire model set-up and parametrization explained for the illustrative riverbank filtration site model holds for the model spin-up and the simulated 20d flood event with a simultaneous increase of microbes and 4He in the river water. To improve clarity for the reader, we have

slightly revised the structure of the model setup section. The initial conditions and model spin-up are now described first, followed by the explanation of the flood event (see revised sentence in lines 544–545 of the newly revised manuscript).

Figure 4. Is it possible to present the figures with a logarithmic color map? This will enable a better separation for concentration variation, allowing for the estimation of the model's sensitivity.

Yes, it is possible to visualize the initial concentrations with a logarithmic color map. However, the logarithmic visualization does not increase the separation for concentration variations. Therefore, to facilitate a direct comparison of the values between the main text and the figure, we have chosen to retain the non-logarithmic color scale.

Line 439: In a way, this sentence: "The simulated concentrations show faster transport of microbes due to size exclusion compared to the slower bulk transport of solutes like 4He," could have easily been written for colloidal transport, and if we focus only on the velocity aspect, then it could be written for sorption scenarios. I believe that an effort should be made to explain how the behavior of bacteria in the model's results differs from that of colloids or solute sorption. The implementation of the bacterial aspect in the model is presented, as is the specific implementation; however, the variation in these aspects on the results from colloid transport with sorption is not discussed. The only aspect I notice is the decay rate, but it is not sufficiently stated in the context of the results.

We agree with the reviewer that microbial transport behavior closely resembles that of colloids, as microbes can be considered reactive bio-colloids. However, in our view, the sentence following the cited reference highlights an important distinction, namely, microbial inactivation processes (lines 567–568 in the previously revised manuscript). While both colloids and solutes can be subject to attachment and detachment processes, the key difference lies in their size. Due to their larger size, colloids (and thus microbes) can become physically excluded from smaller pore spaces, a process known as size exclusion. This mechanism is explained in the introduction (lines 44-45 of the previously revised manuscript) and is also explicitly mentioned in the illustrative example as a distinguishing feature of microbial transport when compared to solute transport (see line 565-567 of the previously revised manuscript). To improve clarity for the reader, we have now added a note on size exclusion when presenting the results before (lines 569-571 of the newly revised manuscript).

Figure 5. Add the "Days" as a label to the x-axis

We assume that the reviewer refers to our initial, unrevised manuscript in which the label of the x-axis of Fig. 5 was indeed missing. In the first revision round, we updated the Figure (Fig. 7 in the revised manuscript) by adding the missing label.

**2. Reply to comments of anonymous reviewer #3**

This manuscript is the revision of a manuscript that I have reviewed before on implementing a dual-permeability approach to address transport of (microbial) colloids in the proprietary software HydroGeoSphere that can simulate saturated-unsaturated flow and transport coupled to surface flow and transport. Upon the revision, the authors have added a new demonstration case that extends the Abdul (1985) benchmark of HydroGeoSphere, which includes coupled surface runoff and subsurface flow and transport, to include transport of microbial colloids. Such a testcase was requested by both reviewers of the original manuscript.

We thank the reviewer for the time and effort to review our manuscript a second time.

However, I am still not convinced. In my previous review I have explained that the dual-permeability model was originally developed for completely different questions, namely to address flow and transport in highly heterogeneous media (including karst systems) by an effective model that does not resolve the heterogeneity. The dual-permeability specific coefficients (second permeability, division of the pore space, exchange coefficient) are typically calibrated by fitting anomalous behavior in both flow (e.g., different head responses in nearby piezometers connected to karst features and not) and transport (e.g., double peaks, extended tailing). None of that applies to the examples of the authors, and it was obviously also not the intension of Bradford et al. (2009), who reinterpreted the approach to address size-exclusion of colloids by introduction of a second permeability. I find it odd that the manuscript contains zero critical discussion of the underlying conceptual assumptions, even after my review (, which I don't want to reiterate). The literature is full of dual-permeability papers that are entirely differently motivated, and the authors simply ignore.

We agree with the reviewer about the initial motivation behind the development of the dual-permeability concept for systems with differences in permeability (e.g. fractured rock with a permeable matrix, near-surface soils containing macropores). The reviewer's reservations are perhaps linked to the use of the term dual-permeability to describe the system we are modelling. Compared to the "classical" dual-permeability systems, it is indeed different because it is really a two-region system that specifically applies to colloids. We retained the term dual-permeability because it was initially proposed by Bradford et al. (2009). Here is the beginning of their abstract:

Recent experimental and theoretical work has demonstrated that pore space geometry and hydrodynamics can play an important role in colloid retention under unfavorable attachment conditions. Conceptual models that only consider the average pore water velocity and a single attachment rate coefficient are therefore not always adequate to describe colloid retention processes, which frequently produce nonexponential profiles of retained colloids with distance. In this work, we highlight a dual-permeability model formulation that can be used to account for enhanced colloid retention in low-velocity regions of the pore space.

They mention using a dual-permeability formulation to account for a high-velocity and a low-velocity region for colloid attachment because using the average pore velocity (with an equivalent porous medium formulation) was found to not always be adequate. They should have perhaps labelled their formulation as two-regions, or two-zone (which we mention in line 78 of the original manuscript) and specify that it is mathematically analogous to a dual-permeability formulation, but not physically similar to systems that are modeled with dualpermeability. We clarified this in lines 88-90 of the newly revised manuscript, where we now state:

"While the mathematical formulation remains similar to its traditional use, the reinterpretation addresses a different physical system in which colloid attachment is influenced by small-scale velocity variations within the porous medium."

I have expressed my skepticism that the dual-permeability approach is really needed in my previous review. Unfortunately, the demonstration cases of the authors don't show that the results obtained could not be obtained by a traditional particle-transport code with single permeability, attachment, detachment, and straining. The authors did not make any effort to make this comparison. They only show that the computationally very demanding and difficult to calibrate dual-permeability approach has successfully been implemented. But it is only an improvement when it leads to a qualitative difference to existing approaches. (A hardly detectable peak in a breakthrough curve does not convince me that the effort is worthwhile.) Thus, I stick to my recommendation to go without the scientifically boring 1-D verification against an analytical solution in the main article (it can be moved to supplementary material and briefly mentioned in the main text) and perform a real effort to reproduce the same results (e.g., of the Abdul test case) with a single-permeability model and the usual parameterizations for particle transport. Please demonstrate that there is a real difference!

The motivation for the work was to develop a model that represents as many physical processes as possible for future applications, such as the datasets of microbes and solute tracers we are in the process of producing and publishing. We would not have developed the model if we had initially thought that a particle tracking model based a single permeability would cover all cases. Our motivation builds on several existing studies and reviews on the matter, as referenced already starting with our original manuscript (e.g., Tufenkji, 2007; Bradford et al., 2014; Molnar et al., 2015). We believe that comparing simplified models such as suggested is outside the scope of the paper but it is certainly something that could be investigated later because we do not advocate to always use the most complex and computationally demanding model available. However, one benefit of developing a more complex model, as we have done here, is that it allows to generate a reference case to compare simplified approaches.

We still consider a verification example for a new code feature an absolute must in the main manuscript of a technical note, and therefore we are not going to remove the comparison between our implementation and the analytical equation. The appreciation of the verification example depends on the personal perspective of the reviewer, as apparent by the contrary comments of the different reviewers.

I also believe that the authors should discuss the issue of calibration in a more systematic way. While it is true that any complex model is difficult to calibrate, the introduction of the second permeability plus fractions and exchange terms in a setting that does not show anomalous flow and solute-transport behavior is an add-on to the already existing complexity. The fact that the authors use exactly the same boundary conditions in both domains might help to some extent. Unlike in karst-applications of the dual-permeability domain, the velocity fields in the two domains are doomed to be identical to a fixed factor and the heads are actually completely identical in the way the authors handle the model, at least in the saturated part of the subsurface domain. The factor of the velocities is the ratio of conductivities, and the effective single conductivity of the dual-permeability model is the fraction-weighted arithmetic average. This might be seen as a chance, as a single-domain flow calibration might

be a good starting point. To get the ratio of the two conductivities and the fractions of the two domains, and the exchange coefficient for transport (that for flow should be insensitive with identical head fields) requires solute-transport information sensitive to that [which I have not seen in the applications of the authors], and the coefficients for attachment, detachment, and straining from particle measurements. The chances are super high that model calibration is very ambiguous, and I honestly don't see how the authors could obtain crucial model parameters independently from concentration data.

The reviewer is correct that the calibration of such models requires not only hydraulic information, but also solute concentration data to derive the ADE related model parameters and resolve for possible spatial heterogeneity. We have recently published a review on the matter of calibrating integrated models with different data types, therefore we are highly aware of these aspects and consider ourselves experts in the domain (Schilling et al., 2019, doi: 10.1029/2018RG000619; and Schilling et al., 2022, doi: 10.1029/2022GL098944). We must point out that the reviewer is not correct that our implementation adds on top of the complexity compared to a normal dual permeability implementation for matrix+fracture transport, because as elaborated at length in our manuscript and as pointed out correctly by the reviewer before, here the dual permeability approach is not used to implement dualpermeability transport that affects solutes, but exclusively to address microbial transport behavior. Thus, anomalous flow behavior that would be observable in solutes is neither expected in cases where this approach is applied for this purpose exclusively, nor is anomalous flow behavior required to calibrate the model. However, alongside solute concentration observations, models based on our implementation of course also require observation of microbial concentrations, as only these alongside solute concentration observations will allow to resolve the parametrization of the dual permeability approach for microbial transport. This we stated clearly in the discussion of both the original manuscript (lines 460-462), as well as in the previously revised manuscript (lines 587-589). The reviewer is correct that the chances are high that observations of just one or two solutes and "one" microbial species could lead to very ambiguous results in calibration. This aspect is a general, well studied problem of inverse modelling in integrated surface-subsurface contexts (as pointed out above, see for example Schilling et al., 2019, doi: 10.1029/2018RG000619), and any analysis of it is outside of the scope of this manuscript. However, to address the reviewer's point, we now describe more clearly that we need solutes and microbial tracer data - ideally of multiple different microbes - where due to the different sizes of the solutes and the different microbes, the inverse problem is sensitive to the fraction of the two regions and the respective conductivities (see lines 608-10 of the newly revised manuscript). As already mentioned in the discussion, with online noble gas and flow cytometry analysis, we are now able to obtain temporally and spatially highly resolved concentration observation time series of multiple solutes and multiple groups of microbes (see lines 610-615 of the newly revised manuscript).

**Minor stuff**

1. Line 83: At the end of the traditional approaches, you should at least mention what is not possible in these approaches.

We added a sentence in the revised manuscript (lines 83-84 of the newly revised manuscript).

2. Somewhere where you discuss the Bradford (2009) model: Explain where the dual-permeability model really comes from, and how Bradford reinterpreted it.

We included a brief explanation in the introduction (lines 85-88 in the newly revised manuscript).

3. Lines 86-87: "... to occur in the small-pore, low velocity regions ... occurs in the large-pore, high-velocity region". It's not the region that is small.

The reviewer is correct. We changed it to "small-pore" to make it clear that it is not the region but the pore space that we refer to.

4. Line 119: Helium is not a radioactive compound. It is the product of alpha-decay, but you simply treat it as conservative compound.

We thank the reviewer for pointing out this typo. 4He is not radioactive but radiogenic (corrected in lines 124 of the newly revised manuscript). Due to its low release rate, it can be treated as a conservative tracer at the considered time scale (days to weeks to a few months) (already mentioned on lines 211-212 in the original manuscript; lines 244-245 of the newly revised manuscript).

5. Equation 12, and lines 185-186: What is the reasoning behind assuming a mass exchange between the solid parts of the high- and low-permeability domains? Is don't see any physical mechanism that could do that.

The mass exchange between the solid phases considers rolling or sliding of microbes on solid surfaces due to hydrodynamic forces (e.g. Bradford et al., 2009; Molnar et al., 2015). We have added a note explaining the physical mechanism in lines 201-202 in the newly revised manuscript.

- 6. Equation 17: the divergence of flux should be written with a nabla operator followed by a scalar-product dot (\cdot in LaTeX) We adjusted this in Equation 17.
- 7. Line 205: The upside-down triangle is the nabla operator, which may represent the gradient (typically of a scalar field) OR the divergence of a vector field. It is not always the gradient.

But in our case, it is the gradient, which is why it is used here in this way.

8. Section 3: see above. Move that to supplementary material.

As it is standard practice to validate new code against an existing model or analytical solution, we consider this an important component of our paper. We also recognize that perspectives may vary, as another reviewer specifically requested a more detailed discussion of the verification example. Hence, we leave this in the main part of the manuscript.

9. Line 261: You don't need a transverse dispersivity in a 1-D problem.

This is correct, we corrected this in the revised manuscript (line 278 and Table 1 of the newly revised manuscript).

10. Line 276: If the first-order decay coefficients are supposed to include straining they should not be identical in both domains.

The purpose of the verification example was to provide a straightforward visual impression that the new feature is correctly coded across several parametrizations rather than mimic exactly microbial transport in a soil column. Moreover, the parametrization with identical first-order decay coefficients was only employed to one out of multiple implementations.

11. Line 288: That the analytical solution contains no colloid exchange between the two solids is not a restriction, as this exchange makes physically no sense to begin with.

The reviewer is not correct. As mentioned above, there are physical processes explaining the colloid exchange between two solid surfaces. We invite the reviewer to interrogate the referenced studies.

12. Line 385 and vey often thereafter: Like in the title, I would remove the word "reactive" when you talk about transport of microbes. There are no chemical reactions involved.

We agree, the term reactive might misleadingly associate microbes undergoing chemical reactions. However, compared to a conservative solute, microbes are considered to be reactive by inactivation and attachment/detachment. For clarification, we replaced reactive by the actual processes, namely inactivation and attachment (line 397 of the newly revised manuscript).

13. Lines 404 and 407: Just talk about "microbes", you don't really consider their species.

As explained in lines 376 of the previously revised manuscript, in this illustrative example we mimic a manure application and the transport of a faecal microbial species. The code is able to handle not just generic microbes, but can be tailored to handle specific microbial species. Therefore, we would like to keep the term microbial species.

14. Lines 408-409: You can be more specific here. I assume that the spots in the stream bed where the microbes pop up are locations with infiltrating conditions. There locations have largely to do with the discretized bathymetry of the channel.

We agree and added a more detailed description and explanation (lines 421-422 of the newly revised manuscript).

15. Lines 422-433, discussion of figure 4: I am pretty sure that that the ratio between highand low-K contributions remains constant because the boundary conditions of the two domains are the same: Qhigh/Qlow = w\_h\*K\_high/(w\_l\*K\_l), which here is simply 10%

This is not entirely correct. With the beginning of stream discharge the contribution from the low-permeability region is 0%, as the fast-flow component arrives earlier. However, within half a day, a constant contribution of 10% is reached.

16. Lines 437 and 444: You faithfully talk about size exclusion, but that is not how the model is really formulated. You have simply an irreversible attachment in the low-K region, which you may use to parameterize the effects that are in reality caused by size exclusion.

The reviewer is correct, we incorporated the effect of size exclusion by an irreversible attachment in the low-permeability region. The conceptualization is not new but taken from an array of previous studies, as outlined previously. We explain this clearly on lines 192-197

of the previously revised manuscript. For this reason, we consider it appropriate to talk here of size exclusion.

17. Line 450: "As second illustrative example for coupled microbial and solute transport"

Thank you for the comment, we corrected this in the revised manuscript (line 463 of the newly revised manuscript).

18. Lines 514-515: Why do you refer to bacteriophages? You don't simulate viruses attacking bacteria.

The reviewer is correct. However, we aimed to base the parametrization of the microbial transport on literature values derived by experiments on the well-field scale (lines 515-516 of the previously revised manuscript). Due to a limitation of experimentally derived rates for prokaryotes with two-site kinetic deposition sites, we considered rates of bacteriophages if no value for prokaryotes was available.

19. Lines 522-523: If the decay coefficient is supposed to represent straining, it should be different in the two domains. You did this better in the Abdul testcase.

As outlined in the Method section, there are two ways to consider straining, either by assuming irreversible attachment in the low-permeability pore space (as done for the Abdul testcase) or by utilizing the first-order sink terms (as done for the Riverbank filtration example).

20. Lines 554-555: Your model formulation does NOT restrict microbial transport to the high-permeability pore space. There is also microbial transport in the low-K region.

The reviewer is correct and this is of course intended; however, the high-permeability region is indeed the dominant flow region for the microbes. We clarified this better in the newly revised manuscript (lines 569-571 of the newly revised manuscript).

21. Lines 564-565: See above. There is no demonstration that the dual-permeability formulation is really needed (and a single-permeability model could not get the job done.)

As previously discussed, studies comparing single- and dual-permeability models for microbial transport already exists and have been reviewed multiple times. Another comparison is not the aim of our manuscript and we instead refer to the extensive literature on the matter.

22. Discussion: See my major remarks above

See our comments above.

23. Lines 589-590: Sorry, but the dual-permeability issues come on top of all the other issues in calibrating surface-water-groundwater model. So this is not an excuse.

This is exactly what we stated in lines 589-590 of the previously revised manuscript: "However, this [required multi-tracer data for robust calibration] is not exclusive to the presented reactive transport model implementation, but is necessary for any surface water-groundwater or larger scale groundwater model (Schilling et al., 2019)."

24. Line 621: The transport is preferential in the high-K region also for solutes. You need the irreversible attachment in the low-K region to make this formulation represent straining.

The reviewer is not correct. As explained in the Method section (lines 192-197 of the previously revised manuscript), straining can be accounted for using either irreversible attachment in the low-permeability region or alternatively an elimination term for the solid phase concentration in the low-permeability region.

25. Lines 624-626: You did not show with your applications that the dual-permeability approach is really needed.

See our comments on the major remarks and 21.